# On Equivariance and Fast Sampling in Video Diffusion Models Trained with Warped Noise

## Abstract

Temporally consistent video-to-video generation is critical for applications such as style transfer and upsampling. In this paper, we provide a theoretical analysis of warped noise—a recently proposed technique for training video diffusion models—and show that pairing it with the standard denoising objective implicitly trains models to be equivariant to spatial transformations of the input noise. We term such models *EquiVDM*. This equivariance enables motion in the input noise to align naturally with motion in the generated video, yielding coherent, high-fidelity outputs without the need for specialized modules or auxiliary losses. A further advantage is sampling efficiency: EquiVDM achieves comparable or superior quality in far fewer sampling steps. When distilled into one-step student models, EquiVDM preserves equivariance and delivers stronger motion controllability and fidelity than distilled non-equivariant baselines. Across benchmarks, EquiVDM consistently outperforms prior methods in motion alignment, temporal consistency, and perceptual quality, while substantially lowering sampling cost.

## 1 Introduction

Video-to-video generative models power a broad spectrum of applications, from sim-to-real transfer and style adaptation to generative rendering and video upsampling. Among these, diffusion-based approaches have emerged as the de facto standard for conditional video generation (Esser et al., 2024; Brooks et al., 2024; Sharma et al., 2024; Agarwal et al., 2025; Blattmann et al., 2023a;b; Peebles & Xie, 2023). Following the original formulations of image and video diffusion models (Peebles & Xie, 2023; Ho et al., 2020; Song et al., 2020; Ho et al., 2022), current methods adopt independent Gaussian noise in their forward process. To enforce temporal consistency, they typically augment the architecture

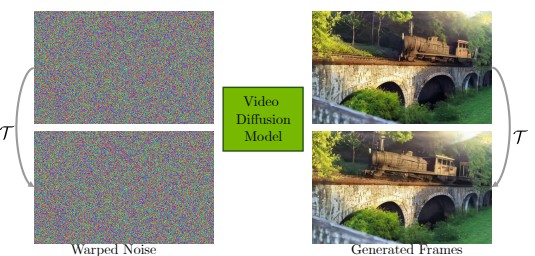

Figure 1: EquiVDM: A video diffusion model that is equivariant to input spatial transformations generates videos with the same spatial transformation when provided with warped noise.

with 3D convolutions (Blattmann et al., 2023a; Yang et al., 2024b) or spatiotemporal attention layers (Peebles & Xie, 2023), enabling stronger propagation of motion information across frames. While these architectural enhancements improve coherence, they often demand large-scale, high-quality video datasets (Agarwal et al., 2025; Chen et al., 2024c) to learn realistic appearance and motion dynamics from unstructured noise.

An alternative line of research seeks to achieve temporal consistency by directly sampling from temporally warped noise. This approach is particularly attractive for video-to-video tasks, where an *input video* naturally provides the motion cues needed to drive the noise warping. In practice, motion vectors (e.g., optical flow) are extracted from the input video and used to correlate Gaussian noise along the motion trajectories. Several works (Chang et al., 2024; Daras et al., 2024; Deng et al., 2024) exploit this idea by warping noise across frames while preserving its spatial Gaussianity, and then applying a pretrained *image* diffusion model to denoise the warped noise, thereby inducing temporally consistent transformations in the output frames. However, as noted by Daras et al. (2024), standard image diffusion networks are not intrinsically equivariant to noise-warping transformations

because of their highly nonlinear layers. As a result, these methods often rely on sampling-time guidance or regularization to approximate equivariance—introducing extra hyperparameters and added complexity. More recently, Burgert et al. (2025b) showed that fine-tuning video diffusion models (VDMs) with warped noise for video-to-video generation tasks can enhance motion control, further underscoring the potential of this direction.

In an era dominated by large video diffusion models trained with independent Gaussian noise, we ask two key questions: (1) What role does warped noise play in training video-to-video diffusion models? and (2) What practical benefits does it provide? We show—both theoretically and empirically—that training with warped noise induces equivariance to spatial warping transformations, a property that emerges simply by replacing independent Gaussian noise with warped noise in the forward process, without altering the conventional VDM objective (see Figure 1). Unlike prior approaches that introduce specialized modules (Khachatryan et al., 2023; Chen et al., 2023a; Zhang et al., 2023b; Lin et al., 2024; Wang et al., 2024; Wu et al., 2024; Karras et al., 2021), our findings reveal that equivariance is an inherent consequence of noise warping. We refer to such models as *EquiVDMs*. Beyond theory, we demonstrate that EquiVDMs generate temporally coherent videos in fewer sampling steps without degrading visual quality. To further accelerate inference, we introduce a distribution-matching distillation method that trains a one-step student EquiVDM with warped noise. This distilled model achieves superior temporal coherence, motion control, and frame quality compared to distilled state-of-the-art baselines trained with independent Gaussian noise. These results are especially significant for real-world applications of video-to-video diffusion, such as sim-to-real transfer, where real-time generation is essential.

In summary, our contributions are: (i) We introduce EquiVDM, a video diffusion model inherently equivariant to spatial warping of the input noise, and show that it can be trained with warped noise using the standard video denoising loss—without requiring any additional regularization. (ii) We demonstrate that EquiVDM produces videos with superior motion fidelity and visual quality compared to state-of-the-art methods. Notably, even the base EquiVDM outperforms existing models that rely on extra modules to encode per-frame dense conditions. Incorporating such control modules (e.g., soft-edge conditions) further improves performance. (iii) We show that EquiVDM with warped noise achieves high-quality video generation in very few sampling steps, enabling faster inference without sacrificing fidelity. (iv) We propose a distribution-matching distillation method that leverages warped noise to train a one-step student EquiVDM for video-to-video generation. We empirically validate that the distilled model preserves equivariance to input warping and delivers stronger motion control and frame quality than distilled state-of-the-art baselines trained with independent Gaussian noise.

## 2 RELATED WORKS

**Controllable video generation** Controllable video generation extends image-generation methods by leveraging additional constraints to guide generation. Prior works incorporate dense frame-wise signals such as depth or edge maps by adding modules to text-to-video backbones or by introducing temporal blocks to capture motion (Chen et al., 2023b; Khachatryan et al., 2023; Lin et al., 2024; Wang et al., 2024). For user-defined sparse trajectories (e.g., drag-and-drop), researchers encode these trajectories through auxiliary modules or flow-completion strategies, then fuse them into the diffusion model's latent features (Li et al., 2024; Yin et al., 2023; Chen et al., 2023a; Wu et al., 2024). Some approaches refine alignment with 2D Gaussian or bounding-box constraints, bypassing the need for an initial frame or applying sampling-time guidance to precisely follow the specified motion (Feng et al., 2025; Namekata et al., 2025).

**Taming noise for rendering and generation** Generating noise with specific properties such as independence and temporal consistency is a crucial step for diffusion model based video generation, as well as rendering in graphics. For example, Wolfe et al. (2021) improve the rendering efficiency and stability by introducing a spatiotemporal noise generation pipeline for stochastic rendering. Kass & Pesare (2011) propose a fast coherent noise generation method for non-photorealistic rendering. Corsini et al. (2012); Goes et al. (2012) focus on 2D blue noise generation for more efficient ray-tracing based rendering pipeline. Huang et al. (2024a) extend the blue noise generation to the diffusion model based video generation given that the blue noise preserves more high-frequency information than Gaussian noise. Ge et al. (2023) study the noise prior and introduce temporally correlated noise in video diffusion without any spatial transformation. Luo et al. (2023); Zhang et al. (2024)

explore the residual noise between frames for video generation with more temporal consistency. In (Lu et al., 2024) the temporal correlation of the noise for video generation is modeled directly to improve temporal consistency.

Getting consistent Gaussian noise for image sequence and video generation using diffusion models has been getting more attention recently. Chang et al. (2024) introduce a warping-based Gaussian noise generation method based on conditional upsampling for image sequence generation. The warped noise theoretically preserves Gaussianity for each frame while being temporally consistent across frames. Deng et al. (2024) improve the efficiency of the warping-based method by operating directly in the continuous domain thus avoiding the need for conditional upsampling. Daras et al. (2024) proposes a consistent Gaussian noise generation method alternatively based on Gaussian process.

Recently, Yan et al. (2025) and Burgert et al. (2025a) utilize the consistent noise for 3D asset and video generation. More specifically, Yan et al. (2025) propose a method for text-to-3D generation by distilling from a pretrained image diffusion model using multi-view consistent noise. Burgert et al. (2025a) finetune a pretrained video diffusion model using warped noise and empirically show that it achieves better motion control. In this work, we theoretically and empirically show that the equivariance to the warping transformation of the input noise can be learned by using the original loss without any modification or new modules. In addition to improved motion controllability, we show that the EquiVDM requires fewer sampling steps, and propose a distillation method using warped noise to train a one-step student for video-to-video generation tasks with superior performance compared to the distilled models without equivariance.

## 3 PRELIMINARY

**Video Diffusion Model.** We represent a video as a sequence of frames $\mathbf{V} = (V^{(0)}, V^{(1)}, \ldots, V^{(K)})$, where $V^{(k)}$ denotes the $k$-th frame. Input conditions such as text prompts or control frames are denoted by $\mathbf{c}$. A video diffusion model $D_\theta(\mathbf{V}_t; \mathbf{c}, t)$ is trained to recover the clean video $\mathbf{V}$ from its noisy counterpart $\mathbf{V}_t = (V^{(0)} + n^{(0)}, V^{(1)} + n^{(1)}, \ldots, V^{(K)} + n^{(K)})$, where $n^{(k)} \sim \mathcal{N}(\mathbf{0}, t\mathbf{I})$ is Gaussian noise added to the $k$-th frame. For brevity, we omit $\mathbf{c}$ and $t$ in $D_\theta(\mathbf{V}_t; \mathbf{c}, t)$ in the following. The model is optimized using the standard denoising loss:

$$\mathcal{L} = \mathbb{E}_{p(t)\,p(\mathbf{V}, \mathbf{v}_t)} \big\| D_\theta(\mathbf{V}_t) - \mathbf{V} \big\|_2^2 = \mathbb{E}_{p(t)\,p(\mathbf{V}, \mathbf{v}_t)} \sum_k \big\| D_\theta^{(k)}(\mathbf{V}_t) - V^{(k)} \big\|_2^2, \tag{1}$$

where $p(t)$ is the noise distribution, and the right-hand side expands the loss across frames. After training, a video can be generated by iteratively denoising samples from Gaussian noise following the sampling schedule.

**Noise Warping.** Successive video frames exhibit temporal consistency, and in visible regions two adjacent frames $V$ and $V'$ can be related by a linear warping operation $V' = \mathcal{T} \circ V$. Recent works extend this idea to the Gaussian noise used in diffusion models. Daras et al. (2024) model noise as a Gaussian process and apply the same warping transformation, $n'_{\text{GP}} = \mathcal{T} \circ n_{\text{GP}}$. Chang et al. (2024) propose an integral formulation that computes warped noise by aggregating deformed pixels from an upsampled noise image. When frame transitions involve only shift or rotation, their noise can similarly be expressed as a linear transformation $n'_{\text{INT}} = \mathcal{T} \circ n_{\text{INT}}$. In this work, we adopt the integral noise formulation of Chang et al. (2024) for warping noise.

## 4 METHOD

In this section, we first introduce EquiVDM, a video diffusion model equivariant to the warping transformations of the input noise. Then we show how to better train EquiVDM to account for the inconsistency in the latent frames obtained from video encoders. Last, we propose a distribution-matching distillation method using warped noise to train a one-step EquiVDM for video-to-video generation tasks.

## 4.1 VIDEO GENERATION WITH TEMPORALLY CONSISTENCY NOISE

Prior works (Chang et al., 2024; Daras et al., 2024) have introduced methods for producing warped noise while preserving per-frame Gaussianity, enabling images to follow the motion patterns of the warped input noise. However, image diffusion models (IDMs) are not inherently equivariant to noise warping, since their generic network layers break this property. As a result, generated sequences often suffer from inconsistencies and abrupt artifacts such as flickering. To mitigate this, Daras et al. (2024) proposed a sampling-time guidance method that regularizes generated pixels using optical flow.

To eliminate the need for such additional regularization or post-training guidance, two strategies are possible: (1) redesign the network architecture to enforce equivariance to input transformations, or (2) learn equivariance directly from training data via tailored losses or training schemes. The first approach requires substantial retraining and and, moreover, constructing equivariant diffusion neural network architectures remains challenging even for simple transformations such as spatial shifts. We therefore adopt the second approach, which avoids architectural modifications and allows efficient finetuning from pretrained models.

Our key result, stated in the following theorem, is that the standard denoising loss in Eq. 1 implicitly trains VDMs to be *equivariant*, provided the input noise is temporally consistent. In other words, no additional losses, hyperparameters, or regularization are required. Training with warped noise alone is sufficient for VDMs to learn equivariance directly from data.

**Theorem 4.1.** *Consider a temporally consistent video with $K$ frames $\mathbf{V} = (V^{(0)}, V^{(1)}, \ldots, V^{(K)})$, where each frame is obtained by a warping transformation of the first frame $V^{(0)}$, i.e., $V^{(k)} = \mathcal{T}_k \circ V^{(0)}$. Let the noisy video $\mathbf{V}_t$ be generated with consistent warped noise $\mathbf{N} = (n^{(0)}, n^{(1)}, \ldots, n^{(K)})$, where $n^{(k)} = \mathcal{T}_k \circ n^{(0)}$. Then the minimizer of the denoising loss in Eq. 1 is a video diffusion model $D_\theta$ that is equivariant to the transformation $\mathcal{T}_k$, i.e., $D_\theta^{(k)}(\mathbf{V}_t) = \mathcal{T}_k \circ D_\theta^{(0)}(\mathbf{V}_t)$, where $D_\theta^{(k)}(\mathbf{V}_t)$ denotes the $k$-th frame of the optimal denoisers output.*

*Proof.* As shown in Vincent (2011), minimizing Eq. 1 with respect to $\theta$ is equivalent to minimizing

$$\mathcal{L} = \mathbb{E}_{p(t)p(\mathbf{V}_t)} \sum_k \left\| D_\theta^{(k)}(\mathbf{V}_t) - \mathbb{E}_{p(\mathbf{V}|\mathbf{V}_t)}\left[ V^{(k)} \right] \right\|_2^2. \tag{2}$$

Using the warping relation $V^{(k)} = \mathcal{T}_k \circ V^{(0)}$, we obtain

$$\mathbb{E}_{p(\mathbf{V}|\mathbf{V}_t)}[V^{(k)}] = \mathbb{E}_{p(\mathbf{V}|\mathbf{V}_t)}[\mathcal{T}_k \circ V^{(0)}] = \mathcal{T}_k \circ \mathbb{E}_{p(\mathbf{V}|\mathbf{V}_t)}[V^{(0)}], \tag{3}$$

where the last equality follows from the linearity of both expectation and the warping operator. Substituting this expression into Eq. 2 gives

$$\mathcal{L} = \mathbb{E}_{p(t)p(\mathbf{V}_t)} \sum_k \left\| D_\theta^{(k)}(\mathbf{V}_t) - \mathcal{T}_k \circ \mathbb{E}_{p(\mathbf{V}|\mathbf{V}_t)}\left[ V^{(0)} \right] \right\|_2^2. \tag{4}$$

This loss is minimized when $D_\theta^{(0)}(\mathbf{V}_t) = \mathbb{E}_{p(\mathbf{V}|\mathbf{V}_t)}\left[ V^{(0)} \right]$ for the first frame, and $D_\theta^{(k)}(\mathbf{V}_t) = \mathcal{T}_k \circ D_\theta^{(0)}(\mathbf{V}_t)$ for all subsequent frames, establishing equivariance. $\square$

The theorem has two key practical implications. First, it shows that when video diffusion models are trained with warped noise, the optimal denoiser becomes equivariant to the warping transformations present in the input noise and any additional video conditioning—without requiring any modification to the training objective. Second, this equivariance implies that if the input noise is warped according to the motion in a video, a VDM trained on such noise will transfer the same motion into its outputs, thereby promoting motion alignment between input and output.

These insights lead to a simple recipe for training EquiVDM: we retain the standard denoising loss in Eq. 1, but construct the noise by warping the first-frame noise along motion vectors extracted from a driving video.

## 4.2 INDEPENDENT NOISE ADDITION

Although the theory suggests that training VDMs with warped noise encourages equivariance to spatial warping in the input, our experiments reveal that such models can still struggle to generate high-quality videos in practice. We hypothesize that several factors break the theoretical assumptions: (1) errors in optical flow lead to inaccuracies in the estimated warping transformations; (2) successive frames in natural videos do not exhibit perfect one-to-one mappings, since camera motion can introduce occlusions and newly visible regions; and (3) while optical flow is estimated in pixel space, most VDMs operate in a latent space produced by encoders that are not guaranteed to be equivariant (Kouzelis et al., 2025).

To better understand this issue, Figure 2 examines the effect of applying warped noise to a latent encoding of a video. We track three pixels across frames and compare their values in the RGB, latent, and corresponding noise spaces. By construction of the warped noise, the RGB and noise values remain consistent across frames, but in the latent space (middle figure) we observe substantial temporal variation. This suggests that latent embeddings of tracked pixels contain additional high-frequency fluctuations that are not captured when simply adding constant warped noise. Since diffusion models rely on a forward process that destroys information across all frequencies to enable reconstruction in the reverse process Kreis et al. (2022); Rissanen et al. (2022), this mismatch undermines the effectiveness of warped noise in latent space.

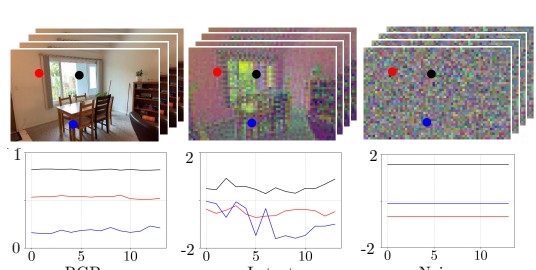

To address this issue, we propose adding a small amount of independent noise to each frame in addition to the temporally warped noise during training. Formally, the injected noise is defined as

$$n = \beta n_{\text{warp}} + \sqrt{1 - \beta^2}\, n_{\text{ind}}, \qquad (5)$$

where $\beta \in [0, 1]$ controls the relative strength of the warped noise, and $n_{\text{ind}}$ denotes independent Gaussian noise.

From another perspective, the added independent component expands the manifold of the noise distribution, enabling it to better cover and disrupt the latent encoding. In contrast, warped noise alone spans a narrower manifold due to strong temporal correlations. Unless otherwise

Figure 2: The values of three tracked points in the video frames in the pixel, latent and noise videos. The variation in the latent video is much larger than the one in the pixel and noise videos due to the compression in the latent space.

specified, we set $\beta = 0.9$ in all experiments, corresponding to injecting only a small fraction of independent noise.

## 4.3 ONE-STEP DISTILLATION MADE EASY

Since EquiVDM enforces the input noise and the generated content to follow the same motion pattern, the input noise and output video are naturally aligned in terms of motion. In Sec. 5.3, we further show that this alignment yields smoother sampling trajectories that are easier to simulate numerically in just a few steps. Building on this observation, we propose a distribution-matching distillation (DMD) method to train a one-step EquiVDM for video-to-video generation tasks. Specifically, the one-step generator (student) $G_\theta$ is trained by minimizing the expectation over $t$ of the KL-divergence between the diffused target distribution $p_{\text{teacher},t}$ from the teacher genarator, and the diffused generated distribution $p_{\text{student},t}$ from the student generator. $G_\theta$ is trained using the gradient (Yin et al., 2024):

$$\nabla_\theta \mathcal{L}_{\text{DMD}} = \mathbb{E}_t\Big(\nabla_\theta \operatorname{KL}\big(p_{\text{teacher},t} \,\|\, p_{\text{student},t}\big)\Big) = -\mathbb{E}_{t,N}\left[\frac{1}{t^2}\int \big(D_{\text{teacher}}\left(G_\theta(N) + tN_s\right) - D_{\text{student}}\left(G_\theta(N) + tN_s\right)\big)\frac{dG_\theta(N)}{d\theta}\right],$$

where $D_{\text{teacher}}$ and $D_{\text{student}}$ are the teacher and student score functions, respectively. During training, the teacher score function is the frozen pretrained EquiVDM finetuned with warped noise as described in Sec. 4.1. For both the one-step student and the student score function, we initialize the networks with the same pretrained EquiVDM as the teacher, and optimized their weights during training. The one-step student $G_\theta$ takes warped noise $N$ as input. The generated video is then diffused with noise $N_s$ warped using the same transformations as $N$. Using identical warping operations for both the student generator and score function not only preserves the equivariance of the student model but also simplifies score estimation for distribution matching.

## 5 EXPERIMENTS

In this section, we first validate the effectiveness of the warped noise input and the learned noise warping equivariance by comparing EquiVDM with other methods without warped noise input. Then we show that EquiVDM generates temporally coherent videos in fewer sampling steps without compromising the quality, and further demonstrate that it leads to one-step distilled model with comparable performance to the multi-step baselines. Last, we perform ablation studies to investigate the effects of the added noise amount and varying sampling steps.

### 5.1 EXPERIMENT SETUP

**Datasets and Metrics**   We curate our dataset for training from the training set of RealEstate-10k  (Zhou et al., 2018), OpenVideo-1M  (Nan et al., 2024) and VidGen-1M (Tan et al., 2024) datasets. The RealEstate10K dataset contains about 80k videos of static real estate scenes, while OpenVideo-1M and VidGen-1M each contains around 1M in-the-wild videos including both static and dynamic scenes. For evaluation, we use Youtube-VIS 2021 (Yang et al., 2019) and MSRVTT (Xu et al., 2016) datasets. We use LLaVA-NeXT (Liu et al., 2024a) for video captioning for datasets without captions. For efficiency, we extract the video captions for every 10 frames assuming that the videos are temporally consistent and the contents do not change too much.

We evaluate video quality using FID (Heusel et al., 2017) and FVD (Unterthiner et al., 2018), and measure alignment with the driving video using the CLIP score (Radford et al., 2021). We also report the UMT score (Liu et al., 2023), which assesses the alignment between the generated video and the input text prompt, and is better at capturing temporal dynamics and motion described in text than the CLIP score.  In addition, we adopt the Image Quality (ImQ), Background Consistency (BgC), and Subject Consistency (SubC) metrics from V-Bench++ (Huang et al., 2024b), which capture per-frame quality (Ke et al., 2021) as well as temporal consistency across frames in the feature space (Radford et al., 2021; Caron et al., 2021). To further assess temporal consistency and motion alignment with ground-truth videos, we extract dense optical flow from the driving video, warp the generated frames accordingly, and compute the cross-frame PSNR (cf-PSNR) between the warped frames and their corresponding targets in the generated sequence.

**Model and Training**   We train EquiVDM by finetuning from the pre-trained VideoCrafter2 (VC2) (Chen et al., 2024b) and VACE-1.3B (Jiang et al., 2025) models with warped noise as the input as describe in Section 4. To adapt VC2 for video-to-video generation, we add and finetune the additional modules from CtrlAdapter (Lin et al., 2024). In particular, we use canny and soft-edge maps extracted from driving videos using (Su et al., 2021) as the control frames for VC2, and depth maps (Yang et al., 2024a) as the control frames for VACE. We use the AdamW optimizer (Loshchilov & Hutter, 2019) with a learning rate of $10^{-4}$ for the finetuning the base model, and $2 \times 10^{-5}$ for the finetuning the added control modules. The model is finetuned on 64 Nvidia A100 GPUs for around 200k iterations.

### 5.2 VIDEO GENERATION

We begin by evaluating whether EquiVDM, trained with warped noise alone in the text-to-video setting—without any additional video conditioning—can outperform models trained with independent noise. Specifically, we test whether warped noise helps the model better capture semantic content and motion alignment. For this experiment, warped noise is generated by extracting optical flow (Teed & Deng, 2020) from the training videos associated with each text prompt. We compare our approach against VDMs with both U-Net and DiT backbones (Zhang et al., 2023a; Jin et al., 2024; Zheng et al., 2024; Yang et al., 2024b; Chen et al., 2024b). The focus of this experiment is to explore whether the noise-equivariance can benefit the video generation without additional modules for input video conditioning. Quantitative results in Table 1 show that higher CLIP scores confirm the noise-equivariant model can infer semantic information directly from warped noise, while cf-PSNR improvements demonstrate that noise-equivariance emerges during training, leading to better alignment between motions in the input noise and the generated videos. The better motion alignment and video quality is a direct result of the learned model equivariance to the warping transformation.

We then evaluate our method on video-to-video generation task. We compare our method against models with additional control modules (Chen et al., 2023b; Khachatryan et al., 2023; Lin et al., 2024; Jiang et al., 2025). The qualitative results are shown in Figure 3. VC2-EquiVDM generates videos

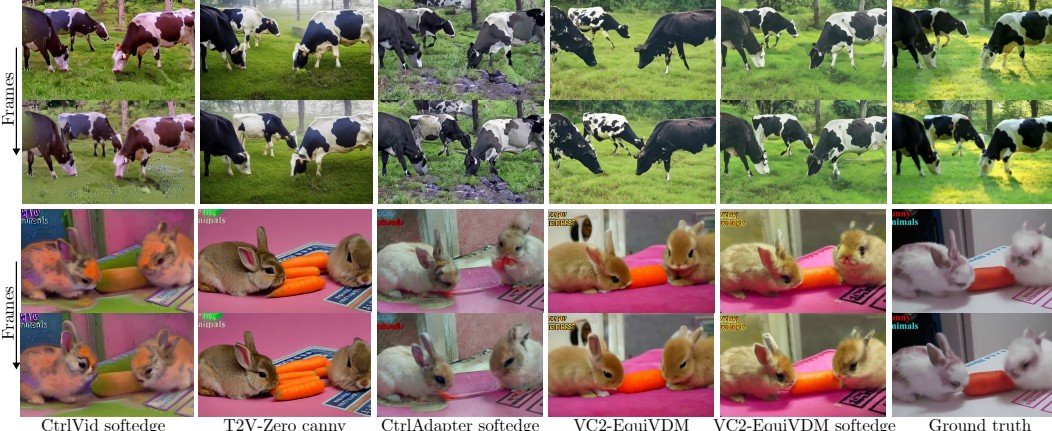

CtrlVid softedge    T2V-Zero canny    CtrlAdapter softedge    VC2-EquiVDM    VC2-EquiVDM softedge    Ground truth

Figure 3: Frames from the generated videos with different video-to-video generation models. VC2-EquiVDM uses warped noise without dense video conditioning; CtrlVid (Chen et al., 2023b), T2V-Zero (Khachatryan et al., 2023), CtrlAdapter (Lin et al., 2024) and VC2-EquiVDM-softedge use either canny edge or softedge.

Table 1: Video generation performance in text-to-video setting without any input video conditioning.

| Method | FID↓ | FVD↓ | CLIP↑ | UMT↑ | cf-PSNR↑ | ImQ↑ | BgC↑ | SubC↑ |
|---|---|---|---|---|---|---|---|---|
| VC2 (Chen et al., 2024b) | 41.23 | 4565 | 0.6500 | 2.6132 | 19.33 | 0.6214 | 0.9250 | 0.9492 |
| Show-1 (Zhang et al., 2023a) | 34.83 | 5422 | 0.6908 | 2.6858 | 20.59 | 0.5633 | 0.9264 | 0.9272 |
| Pyramid-flow (Jin et al., 2024) | 46.88 | 5726 | 0.6377 | 2.5341 | 21.86 | 0.6331 | 0.9446 | 0.9537 |
| OpenSora-1.2 (Zheng et al., 2024) | 39.14 | 5733 | 0.6898 | 2.9288 | 20.35 | 0.6312 | **0.9561** | 0.9740 |
| CogVideoX-2B (Yang et al., 2024b) | 36.76 | 5369 | 0.6540 | **3.0026** | 18.05 | 0.6002 | 0.9490 | 0.9534 |
| VC2-EquiVDM | **26.59** | **3193** | **0.6925** | 2.7487 | **25.65** | **0.6485** | 0.9438 | **0.9747** |

Table 2: Video generation performance in video-to-video setting with input video conditioning.

| Method | FID↓ | FVD↓ | CLIP↑ | UMT↑ | cf-PSNR↑ | ImQ↑ | BgC↑ | SubC↑ |
|---|---|---|---|---|---|---|---|---|
| IntegralNoise canny (Chang et al., 2024) | 39.68 | 3238 | 0.7262 | 2.2782 | 14.09 | 0.5273 | 0.9221 | 0.9345 |
| CtrlVid canny (Chen et al., 2023b) | 38.45 | 2724 | 0.7154 | 1.7470 | 22.68 | 0.6459 | 0.8991 | 0.8622 |
| CtrlVid softedge (Chen et al., 2023b) | 59.80 | 2694 | 0.7129 | 1.3374 | 23.16 | 0.5480 | 0.9051 | 0.8668 |
| T2V-Zero canny (Khachatryan et al., 2023) | 29.98 | 3350 | 0.7146 | 2.5818 | 21.57 | 0.5652 | 0.8736 | 0.8761 |
| CtrlAdapter softedge (Lin et al., 2024) | 39.62 | 2789 | 0.7167 | 2.3857 | 21.52 | **0.6570** | 0.8829 | 0.8683 |
| CtrlAdapter canny (Lin et al., 2024) | 36.24 | 2496 | 0.7214 | 2.3681 | 23.09 | 0.6173 | 0.8773 | 0.8628 |
| VACE depth (Jiang et al., 2025) | 27.04 | 2989 | 0.7421 | 2.7754 | **28.14** | 0.6056 | **0.9448** | 0.9494 |
| VC2-EquiVDM softedge | 24.40 | 2122 | 0.7293 | 2.7267 | 26.86 | 0.5817 | 0.8880 | 0.8764 |
| VC2-EquiVDM canny | **22.24** | **1922** | 0.7551 | 2.7298 | 26.58 | 0.6244 | 0.8787 | 0.8688 |
| VACE-EquiVDM depth | 23.61 | 2329 | **0.7890** | **2.9437** | 29.55 | 0.6503 | 0.9394 | 0.9451 |

from warped noise without using dense conditioning, while other methods use either canny edges or HED softedge Xie & Tu (2015). VC2-EquiVDM-softedge achieves the best temporal consistency for textures (*e.g.* the the patterns of the cows, the grass texture), as well as the motion alignment (*e.g.* the motion of legs of the cows, the orientation of rabbit's head) with the ground truth video where the warping optical flow is extracted from.

The quantitative results are listed in Table 2. Our method achieves the best performance on frame quality, semantic and motion metrics. This manifests that EquiVDM can benefit video generation by taking advantage of the temporal correlation from the warped noise input. It also indicates that the temporal correlation in the warped noise can serve as a strong prior for both the motion pattern and semantic information in addition to motions. We finetuned an image diffusion model (Chang et al., 2024) (IDM) with warped noise to evaluate whether IDM can be trained to be equivariant. The drastic decrease in the cf-PSNR score indicates that the finetuned IDM is not equivariant to the input noise warping without introducing additional regularization or sampling-time guidance.

Another observation is that for our method, the performance of the video-to-video model is generally better than the base model, indicating that the benefit of equivariance is complementary to the

Table 3: Comparison for VACE and VACE-EquiVDM with different sampling steps.

| Steps | FID↓ | FVD↓ | CLIP↑ | UMT↑ | cf-PSNR↑ | ImQ↑ | BgC↑ | SubC↑ |
|---|---|---|---|---|---|---|---|---|
| 10-step VACE (Jiang et al., 2025) | 27.04 | 2989 | 0.7421 | 2.6801 | 28.14 | 0.6056 | 0.9448 | 0.9494 |
| 10-step VACE-EquiVDM | 23.61 | 2329 | 0.7890 | 2.8582 | 29.55 | 0.6503 | 0.9394 | 0.9451 |
| 5-step VACE | 31.63 | 3000 | 0.7214 | 2.5671 | 28.19 | 0.5576 | 0.9421 | 0.9446 |
| 5-step VACE-EquiVDM | 23.95 | 2271 | 0.7894 | 2.6847 | 30.15 | 0.6514 | 0.9361 | 0.9456 |
| 3-step VACE | 38.06 | 3037 | 0.6951 | 2.1759 | 28.64 | 0.5005 | 0.9364 | 0.9354 |
| 3-step VACE-EquiVDM | 26.16 | 2274 | 0.7815 | 2.6166 | 30.63 | 0.6542 | 0.9317 | 0.9422 |
| 1-step distilled VACE | 29.42 | 3004 | 0.7413 | 2.7487 | 28.94 | 0.6367 | 0.9204 | 0.9554 |
| 1-step distilled VACE-EquiVDM | 25.94 | 2553 | 0.7828 | 2.8143 | 29.38 | 0.6520 | 0.9283 | 0.9427 |

additional conditioning modules. As a result, for video-to-video generation tasks, we can improve the performance by making the full model noise-equivariant without any architecture modification to it.

### 5.3 Few-step generation for EquiVDM

To study how the wapred noise input affects the generation, we plot the generation trjaectory curvature for VACE (Jiang et al., 2025) with independent noise and VACE-EquiVDM with warped noise in Figure 4. As shown, the curvature for VACE-EquiVDM is significantly lower compared to VACE with inpendent noise, indicating that the warped noise input helps to make the generation trajectory more straight, hence we can use fewer sampling steps to generate videos with similar or better quality.

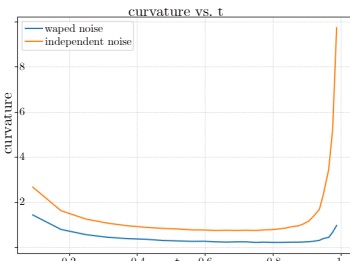

To verify this, we compare the generation performance of VACE and VACE-EquiVDM with different numbers of sampling steps in Table 3. The degradation in quality and semantic alignment for VACE-EquiVDM is much slower compared to VACE with fewer sampling steps. In addition, we use the method described in Section 4.3 to distill the VACE-EquiVDM model into a one-step model, and compare the performance with the one-step VACE-EquiVDM. The distilled one-step VACE-EquiVDM model achieves better performance than the one-step VACE, and matches the performance of the 10-step VACE with independent noise.

Figure 4: Straightness of generation trajectories for VACE (Jiang et al., 2025) with independent noise and VACE-EquiVDM with warped noise.

### 5.4 Ablation Studies

**Sampling steps** Since the motion information about the video is already included in the warped noise, one natural question is whether the number of sampling steps can be reduced compared to the one using independent noise where both the motion and appearance have to be generated from scratch. To answer this question, we first inspect how the warped noise changes the distance between the input noise and corresponding latents of real videos during training. As shown in Figure 5 (a), with the warped noise ($\beta > 0$), the noise-video distance is lower compared to the independent noise ($\beta=0$), and the distance decreases as $\beta$ becomes larger. This indicates that the warped noise is more *aligned* with the target video, and similar to the observation in optimal transport flows (Pooladian et al., 2023; Tong et al., 2023), this data-noise alignment can make sampling trajectories easy to integrate with fewer steps.

We further evaluate our method on ScanNet++ with different numbers of sampling steps using VC2-EquiVDM (Chen et al., 2024b) with soft-edge maps as the control frames (Lin et al., 2024). As shown in Figure 5 (b), with warped noise input, our method can generate videos with similar or better quality compared to the one using independent noise in much fewer sampling steps. In addition, the metrics saturate quickly, indicating that the appearance of the video can be generated from scratch with few sampling steps given the warped noise input. As shown in Figure 6, the detailed appearance-like reflection on the table surface can be generated in as few as 5 sampling steps. These results open up new venues for video diffusion acceleration with warped noise.

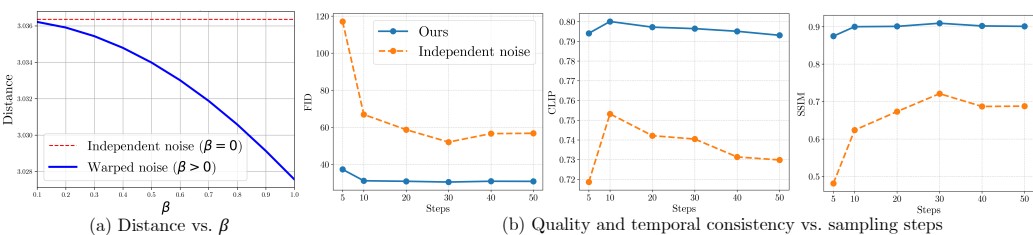

(a) Distance vs. $\beta$

(b) Quality and temporal consistency vs. sampling steps

Figure 5: (a) The noise-to-video distance reduces with the warped noise. (b) Less sampling steps are needed for EquiVDM to achieve similar or better quality compared to independent noise.

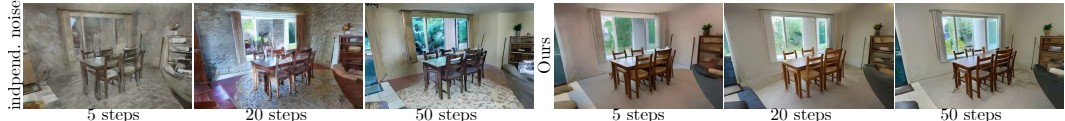

Figure 6: Our method with warped noise generates videos with similar quality compared to the one using independent noise, but with much fewer sampling steps.

**Added noise amount** We evaluate our method with different amounts of added independent noise by adjusting the $\beta$ value in Equation 5. A smaller $\beta$ value indicates more noise added to the video hence less warped noise, and vice versa. In particular, for $\beta = 0.0$ the input noise is independent for each frame without any temporal consistency; while $\beta = 1.0$ indicates the input noise is fully determined by the first frame and the warping operation without any variations.

We evaluate the performance on the test set of RealEstate10K dataset. As shown in Table 4, using warped noise helps in generating better videos in terms of quality, semantic alignment, and temporal consistency. On the other hand, without any added independent noise, the performance degrades since the model fails to model the high-frequency temporal variations of the corresponding pixels in the latent space; while the added independent noise expands the manifold of the input noise such that it covers the

Table 4: Ablations on added noise weight $\beta$.

| $\beta$ value | FID | FVD | CLIP | PSNR | SSIM |
|---|---|---|---|---|---|
| 0.0 | 39.92 | 2292 | 0.8126 | 20.81 | 0.6057 |
| 0.5 | 26.66 | 1765 | 0.8509 | 30.77 | 0.9258 |
| 0.9 | **25.12** | **1585** | 0.8575 | **31.91** | **0.9343** |
| 1.0 | 50.03 | 1910 | **0.9224** | 28.67 | 0.9224 |

latent space better, as discussed in Section 4.2. We found that adding a small amount of independent noise with $\beta = 0.9$ achieves the best balance between quality and consistency.

# 6 LIMITATION AND CONCLUSION

In this work, we introduced EquiVDM, a video diffusion model inherently equivariant to spatial warping of the input noise. We showed that EquiVDM can be trained with warped noise using the standard denoising loss without additional regularization, and that it produces videos with superior motion fidelity and visual quality compared to state-of-the-art methods. We further demonstrated that EquiVDM achieves high-quality video generation in only a few sampling steps, enabling faster inference without sacrificing fidelity. In addition, we proposed a distribution-matching distillation method that leverages warped noise to train a one-step student EquiVDM for video-to-video generation.

Our approach has two main limitations. First, it requires optical flow from the driving video to warp the noise, which is not always available, e.g., in text-to-video generation. A potential remedy is to generate optical flow directly from the text prompt and use it to warp the noise. Second, for long video generation, warped noise input alone does not fully prevent drifting. Future directions include incorporating auto-regressive video diffusion models trained with Diffusion/Self-Forcing (Chen et al., 2024a; Huang et al., 2025) to address this issue.

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

## A   CF-PSNR USED FOR EVALUATING TEMPORAL CONSISTENCY

The cf-PSNR metrics in our paper are used for evaluating the temporal consistency of the generated frames, as well as how the motion pattern of the generated frames follows the optical flow of the input noise. As shown in Figure 7, to compute the cf-PSNR metrics, we first extract the 2D optical flow of the input driving video. Then given the correpsonding generated video, we warp the the source frame (the frame t in the case shown in the illustration) towards the target frame (the frame t+1) using the optical flow. Then we compute the cf-PSNR metrics between the warped source frame and the target frame. As a result, if the generated video follows the same motion pattern as the ground truth and maintains temporal consistency, it will yield a higher cf-PSNR score—and vice versa.

Compared with the metrics in Video Benchmark Liu et al. (2024b), our metric is similar to the "Warping Error" for temporal consistency in the Sec.4.4 of that paper. The only difference is that the optical flow used for warping is estimated from the ground truth video rather than generated video.

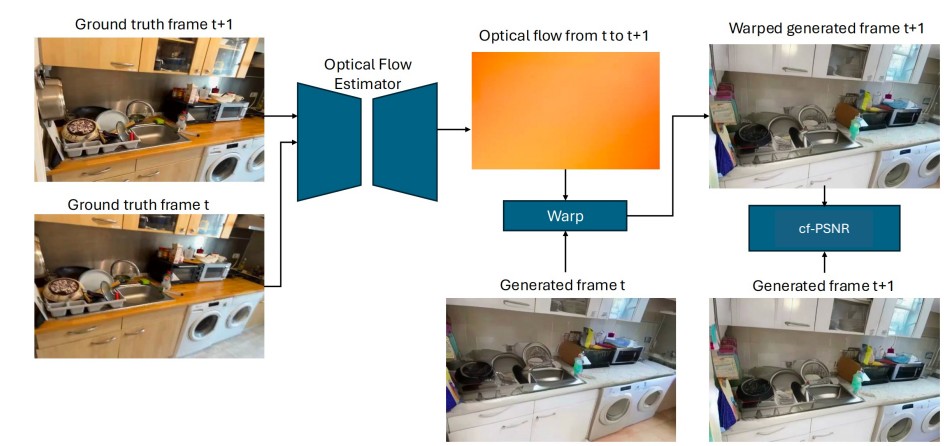

Figure 7: Illustration of the cf-PSNR metrics used for evaluating temporal consistency.

## B   MORE RESULTS ON FEW-STEP VIDEO GENERATION WITH EQUIVDM

In this section, we provide more results on the few-step video generation using VACE-1.3B Jiang et al. (2025) VACE-EquiVDM. Both models use the depth input video conditioning. We first plot the FID scores for the few-step video generation with both models in Figure 8. The EquiVDM model generates better quality videos with fewer steps. Even with 3 steps, the EquiVDM model can generate videos with on-part quality to the VACE-1.3B model at 10 steps.

In addition, the FID score degrages more gracefully for the EquiVDM model. More qualitative results are shown in Figure 9 and the accompanying html file.

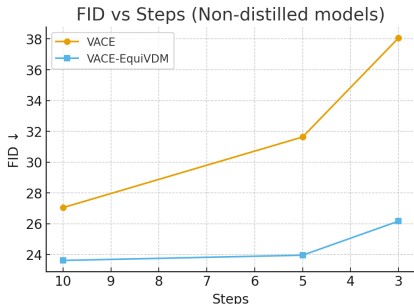

Figure 8: FID scores for the few-step video generation with both models.

## C   EQUIVDM FOR
## DIFFUSION MODELS WITH TRANSFORMERS

For video diffusion models with transformer backbone Peebles & Xie (2023); Yang et al. (2024b); Agarwal et al. (2025), the latent space of the video where the diffusion and sampling process are performed is a set of video tokens from a video tokenizer. Unlike the VAEs in the UNet-based video diffusion models, the video tokenizer not only compress the spatial dimension of the video,

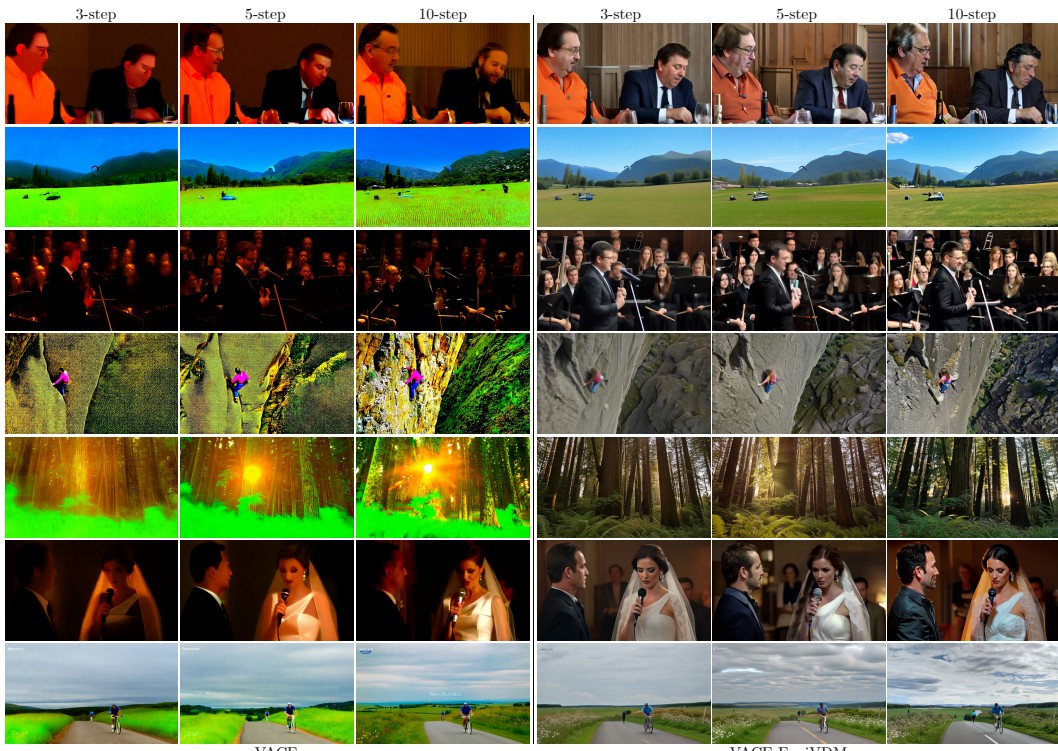

Figure 9: Qualitative results for the few-step video generation for VACE-1.3B and VACE-EquiVDM.

but also the temporal dimension. For example, in CogVideoX Yang et al. (2024b) and CosMos Agarwal et al. (2025), the tokenizer processes a video with $N$ frames by first encoding the initial frame independently. It then encodes the subsequent $N - 1$ frames into a sequence of $\lceil (N - 1)/k \rceil$ temporal tokens, where $k$ represents the temporal compression factor.

We build the warped noise frames accordingly to account for the temporal compression in the video tokenizer. For example, for the video tokenizer temporal compression scheme in CogVideoX Yang et al. (2024b) and CosMos Agarwal et al. (2025), we first get the subsampled video by taking the first frame and every $k$-th frame from the following frames. Then we build the warped noise frames from the subsampled video. Another option is to build the warped noise frames directly from the original video, then subsample the warped noise frames accordingly. The first apporach is more efficient since it reduces the numbers of optical flow estimations. On the other hand, the second approach is more robust to videos with large motions. In our experiment, we use the second apprach for more robustness.

To add the control signal such as soft-edge maps, we use the same method as in the UNet-based video diffusion models: we add the adapter layers Lin et al. (2024) between the frame encoder for the controlling frames and the transformer blocks in the video diffusion model. We interlace the adapter layers every 4 transformer blocks in the transformer backbone to avoid memory overflow. The qualitative results of the EquiVDM with the CogVideoX Yang et al. (2024b) model are shown in Figure 10–13.

## D    ADDITIONAL RESULTS FOR COMPARSIONS WITH OTHER METHODS

In Figure 14-19, we provide additional qualitative results for the comparison in Table 2 in Section 5.2. Please refer to the accompanying html file for the video results.

864
865
866
867
868
869
870
871
872
873
874
875
876
877
878
879
880
881
882
883
884
885
886
887
888
889
890
891
892
893
894
895
896
897
898
899
900
901
902
903
904
905
906
907
908
909
910
911
912
913
914
915
916
917

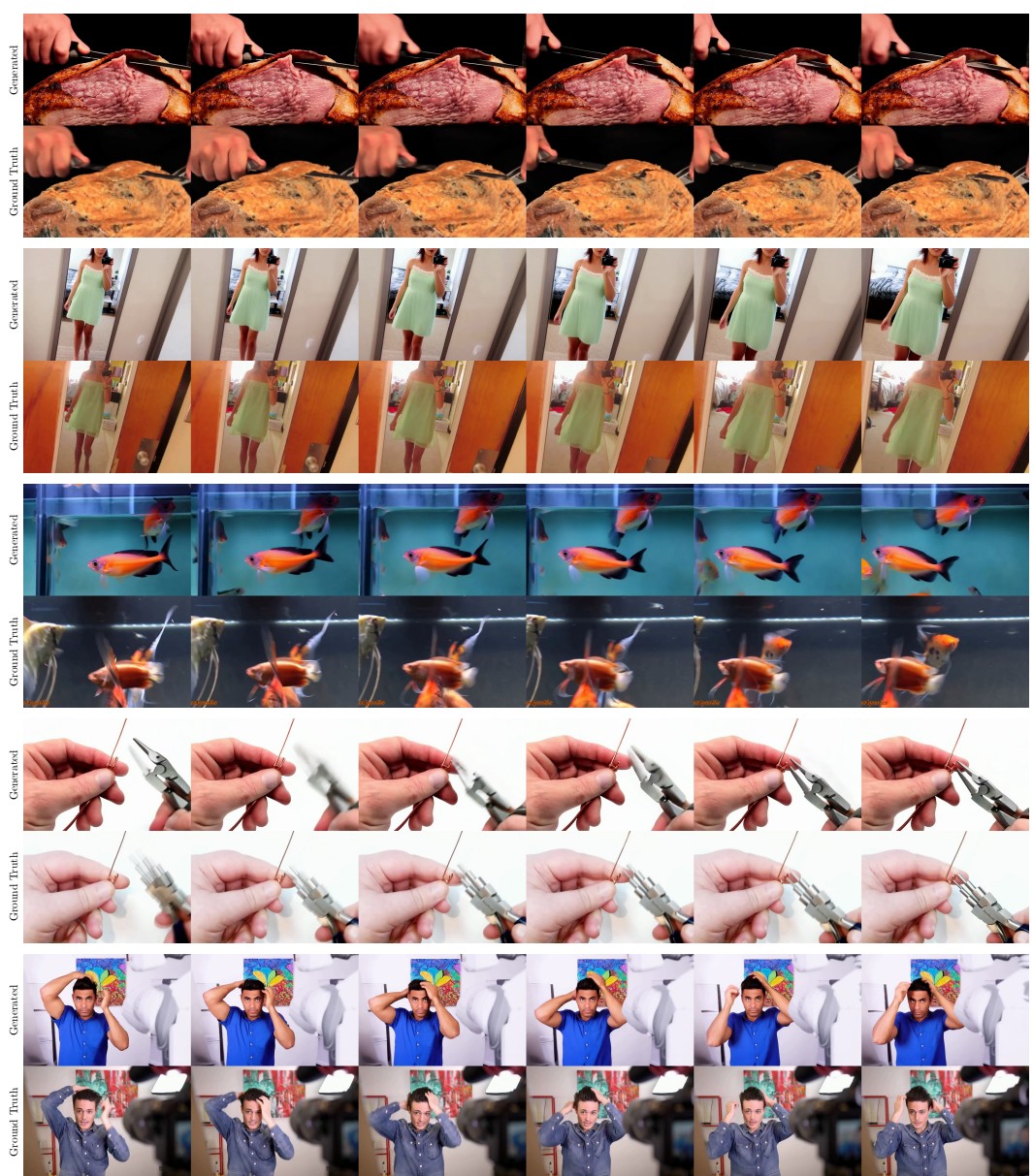

Figure 10: The generated and driving videos of DiT-based video diffusion models.

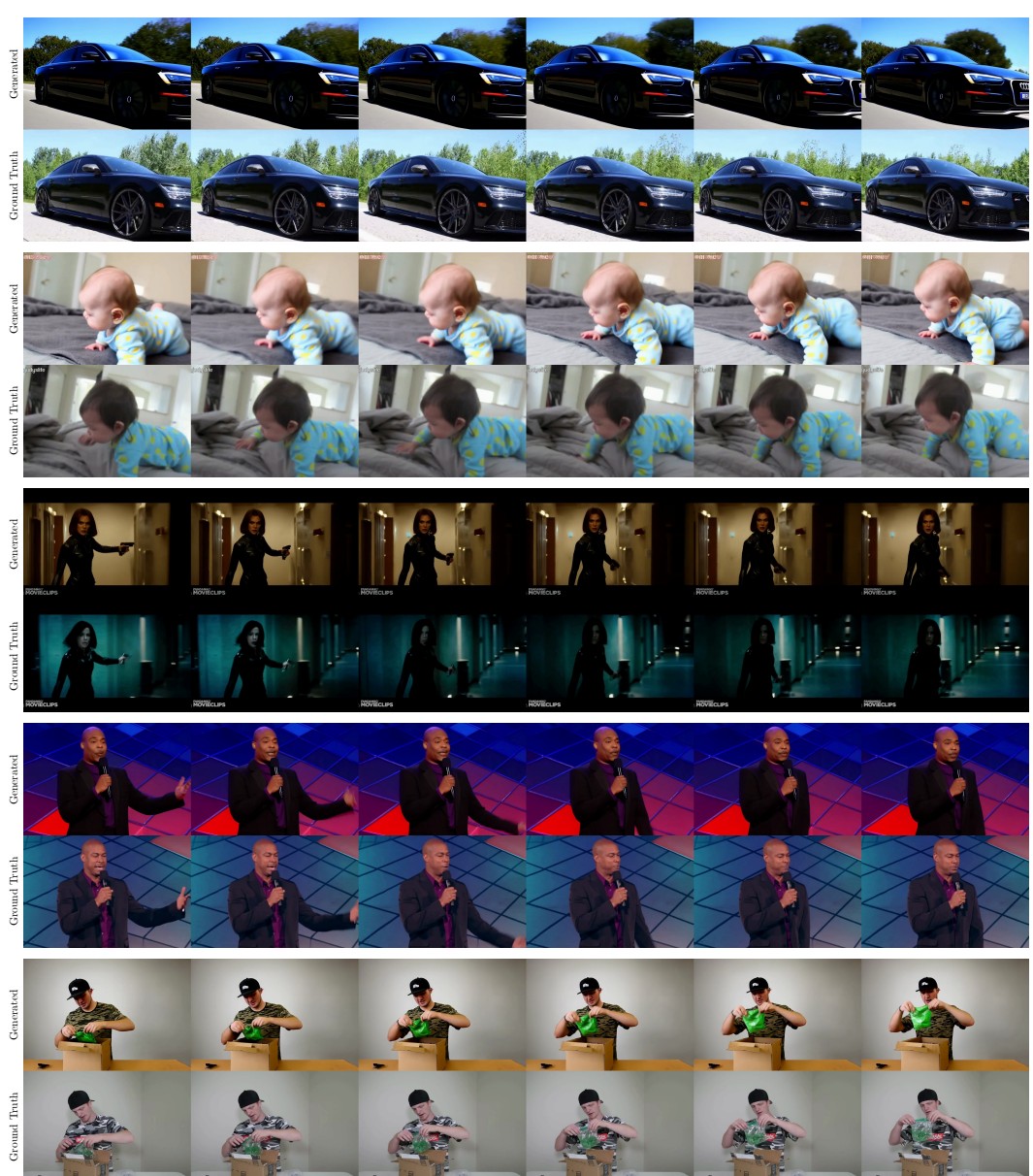

Figure 11: The generated and driving videos of DiT-based video diffusion models.

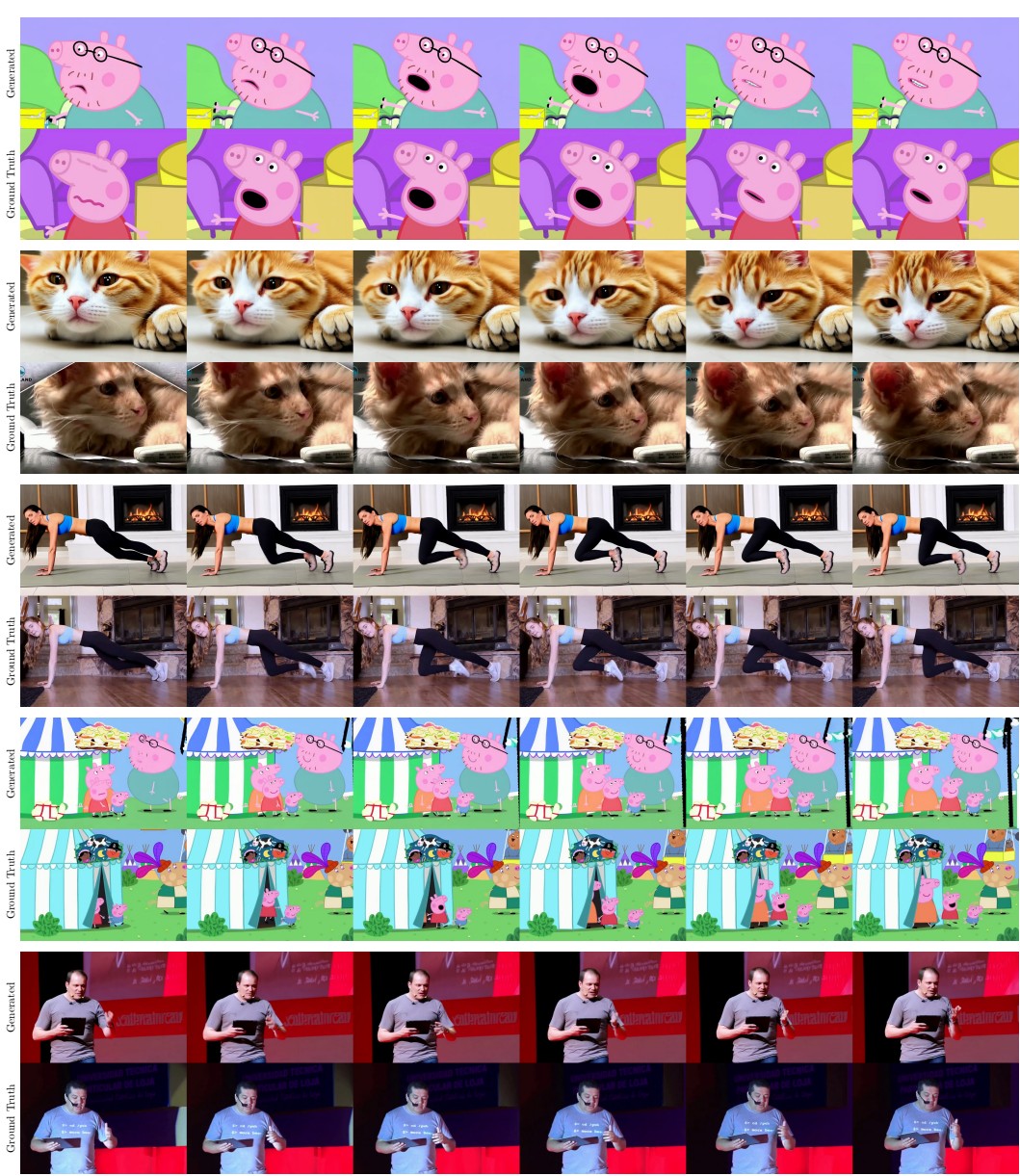

Figure 12: The generated and driving videos of DiT-based video diffusion models.

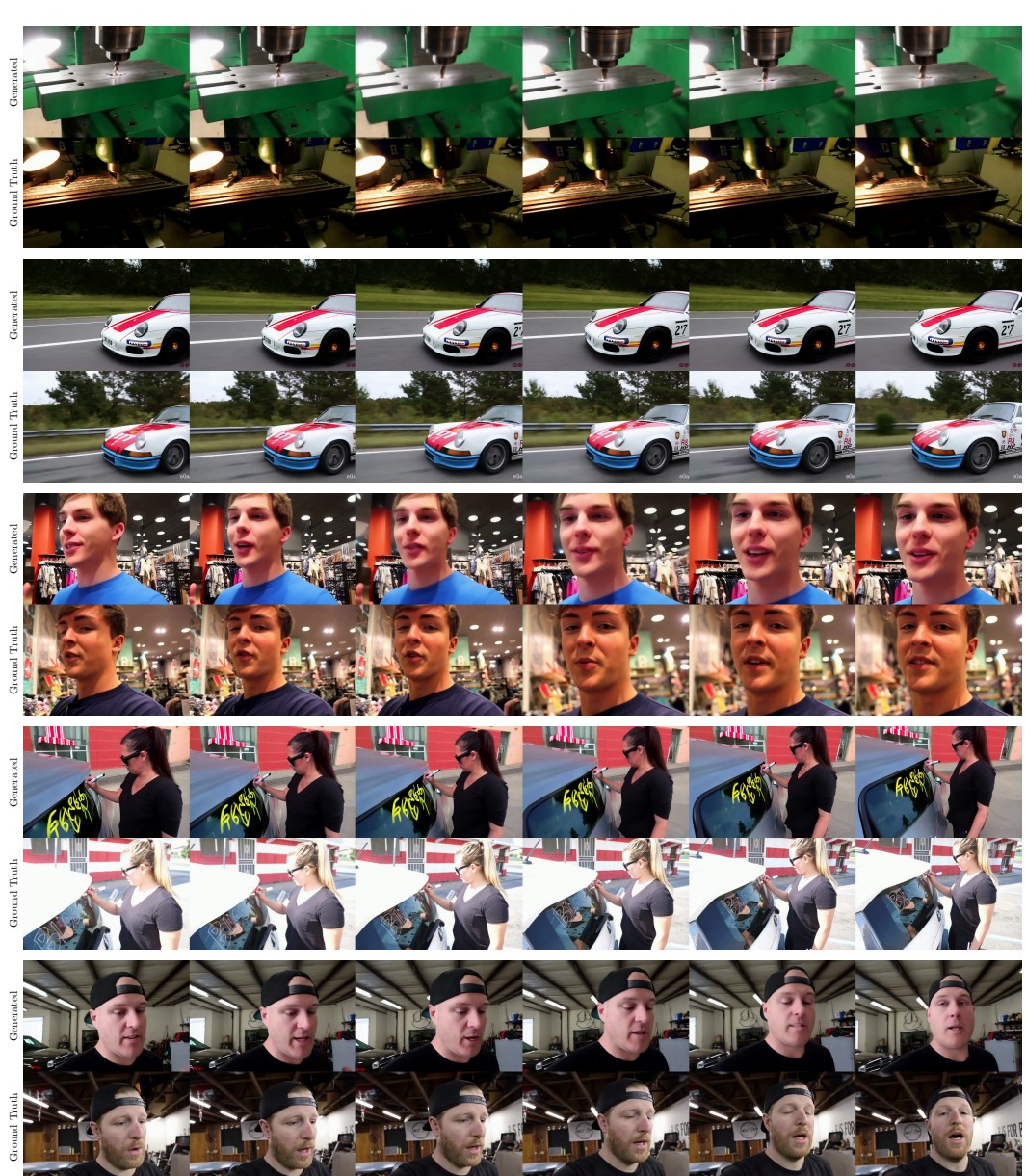

Figure 13: The generated and driving videos of DiT-based video diffusion models.

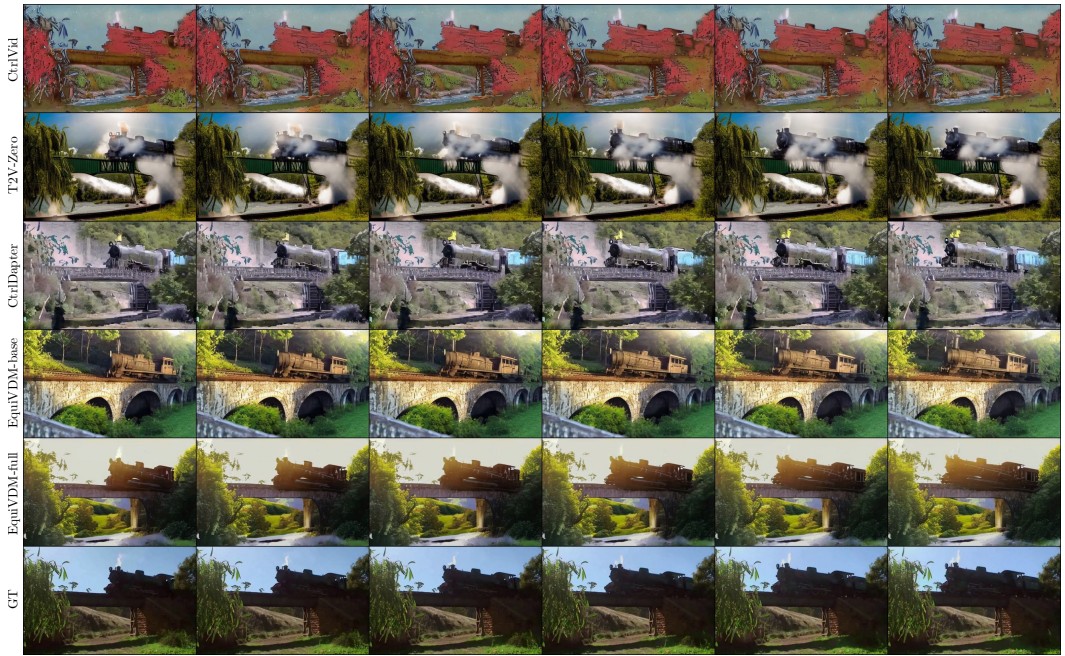

Figure 14: Comparison of EquiVDM with other methods. CtrlVid Chen et al. (2023b), T2V-Zero Khachatryan et al. (2023), CtrlAdapter Lin et al. (2024) and EquiVDM-full used soft-edge map as control signal for each frame along with the text prompt. EquiVDM-base generates videos from warped noise without using dense conditioning.

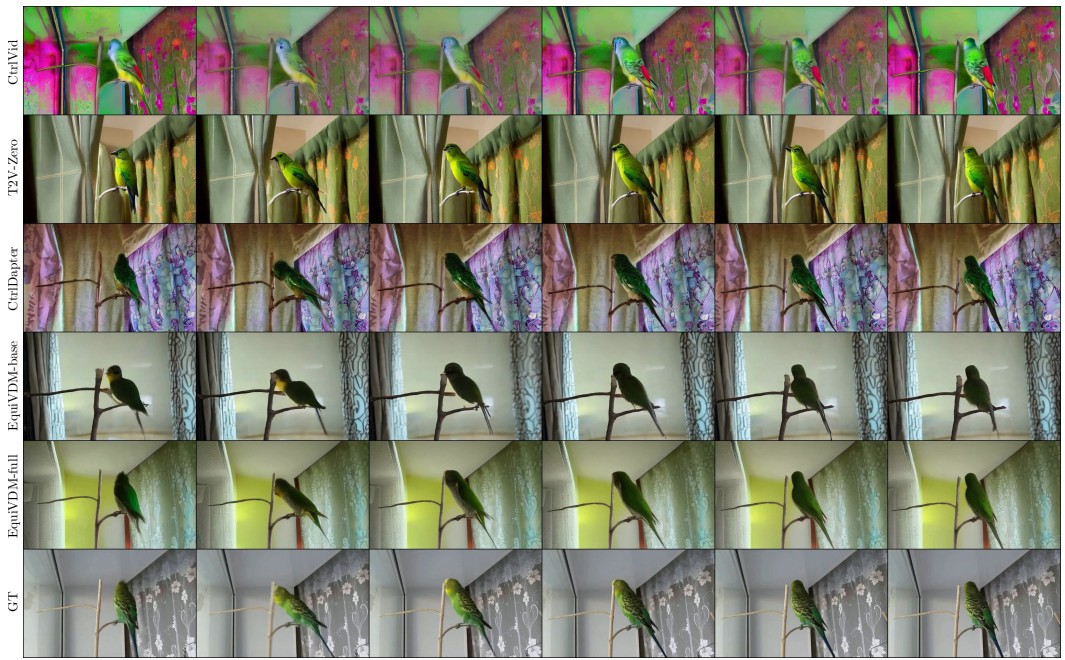

Figure 15: Comparison of EquiVDM with other methods. CtrlVid Chen et al. (2023b), T2V-Zero Khachatryan et al. (2023), CtrlAdapter Lin et al. (2024) and EquiVDM-full used soft-edge map as control signal for each frame along with the text prompt. EquiVDM-base generates videos from warped noise without using dense conditioning.

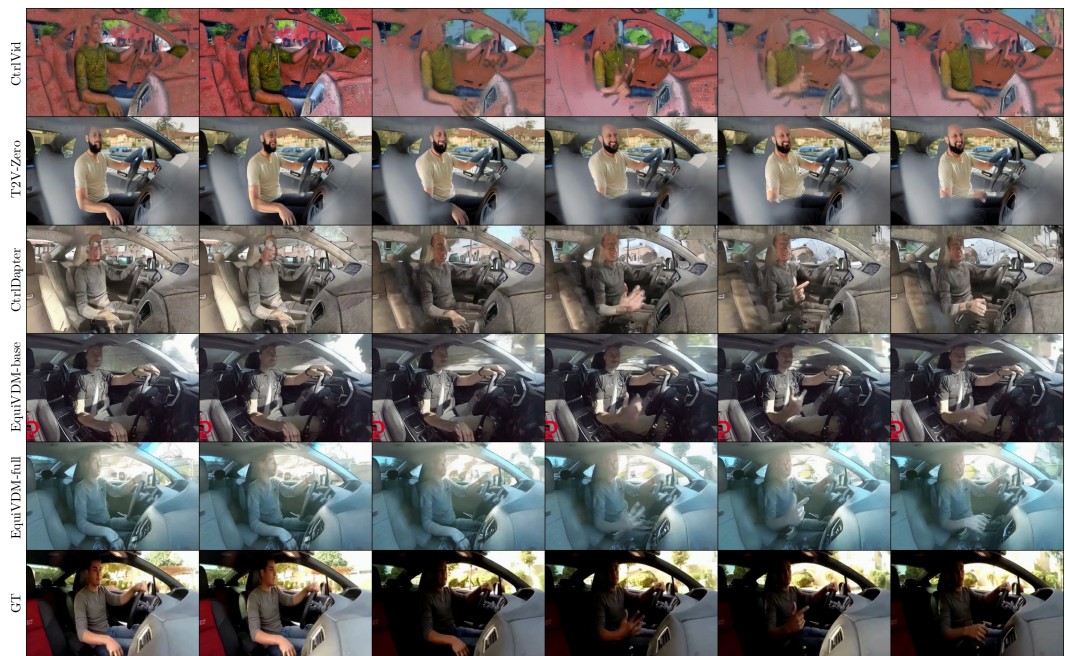

Figure 16: Comparison of EquiVDM with other methods. CtrlVid Chen et al. (2023b), T2V-Zero Khachatryan et al. (2023), CtrlAdapter Lin et al. (2024) and EquiVDM-full used soft-edge map as control signal for each frame along with the text prompt. EquiVDM-base generates videos from warped noise without using dense conditioning.

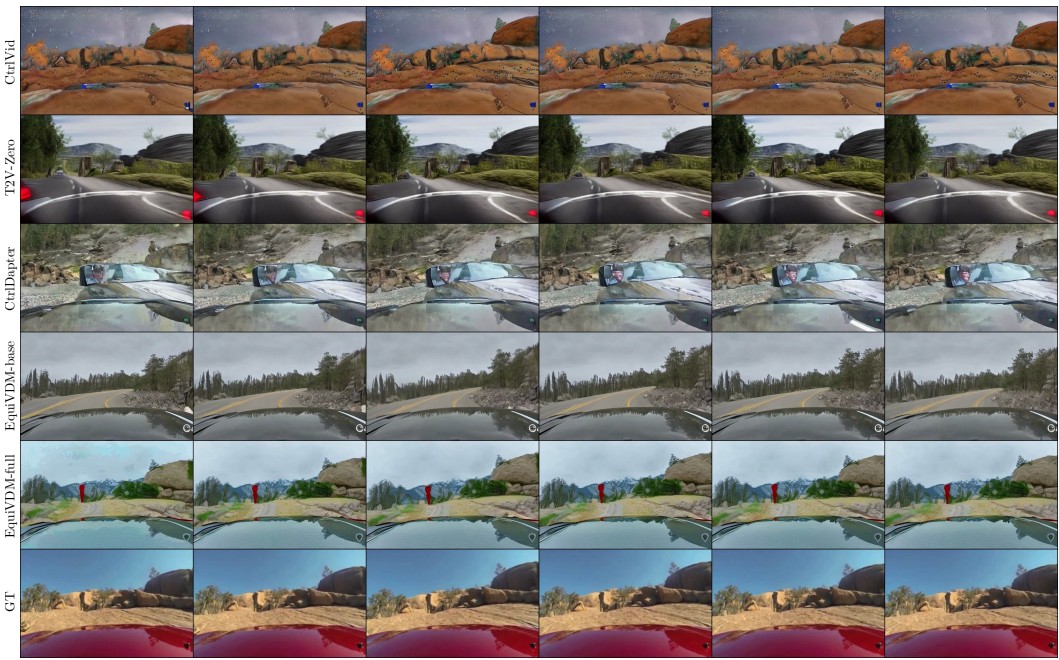

Figure 17: Comparison of EquiVDM with other methods. CtrlVid Chen et al. (2023b), T2V-Zero Khachatryan et al. (2023), CtrlAdapter Lin et al. (2024) and EquiVDM-full used soft-edge map as control signal for each frame along with the text prompt. EquiVDM-base generates videos from warped noise without using dense conditioning.

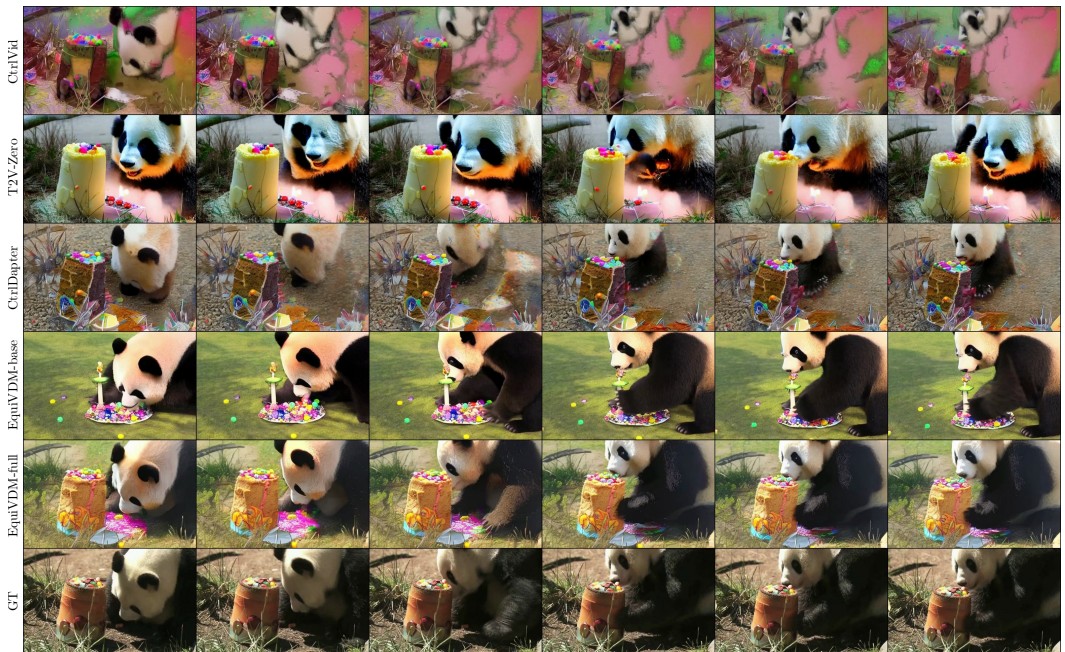

Figure 18: Comparison of EquiVDM with other methods. CtrlVid Chen et al. (2023b), T2V-Zero Khachatryan et al. (2023), CtrlAdapter Lin et al. (2024) and EquiVDM-full used soft-edge map as control signal for each frame along with the text prompt. EquiVDM-base generates videos from warped noise without using dense conditioning.

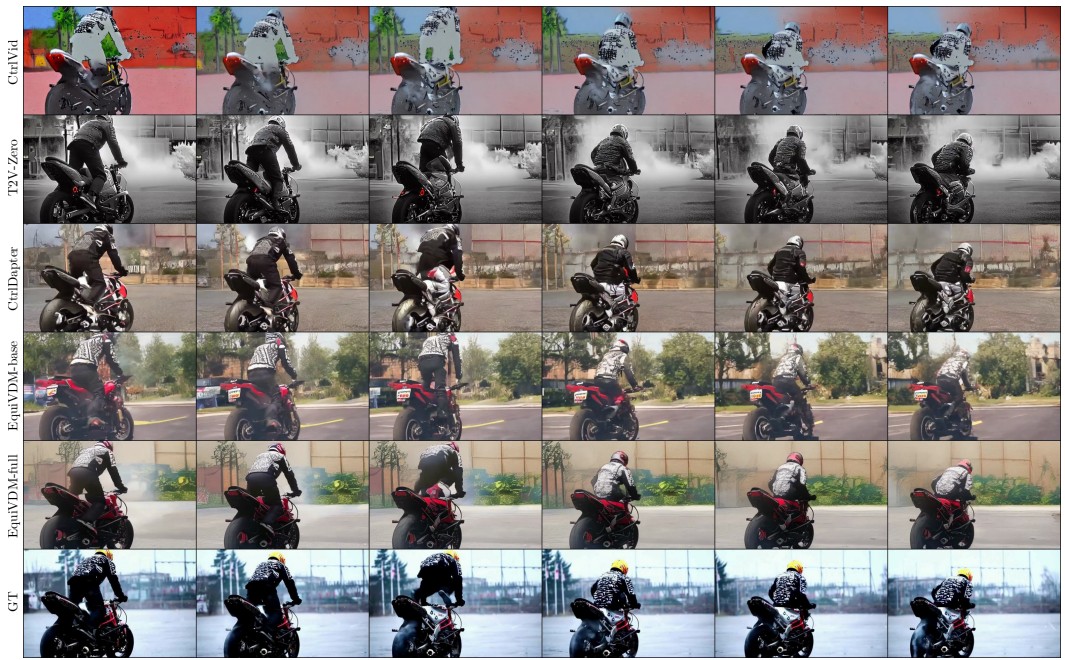

Figure 19: Comparison of EquiVDM with other methods. CtrlVid Chen et al. (2023b), T2V-Zero Khachatryan et al. (2023), CtrlAdapter Lin et al. (2024) and EquiVDM-full used soft-edge map as control signal for each frame along with the text prompt. EquiVDM-base generates videos from warped noise without using dense conditioning.

