# OpenReview forum: "On Equivariance and Fast Sampling in Video Diffusion Models Trained with Warped Noise"
_ICLR.cc/2026/Conference — Submitted to ICLR 2026_

### Official Review · Reviewer_T9uo · 2025-10-29

**Soundness:** 3
**Presentation:** 3
**Contribution:** 2
**Rating:** 4
**Confidence:** 4

**Summary:**

This paper starts from an interesting theoretical observation that training with warped noise iduces equivariance to spatical transformations of the input, but suffers from significant flaws overall. Through the whole paper, I can not find any detail method description, no pseudo-code for this method. Its core theory is built upon warped noise methos proposed by others (see Line 152), and its main technical contribution (one-step distillation) utilizes a method similar to existing DMD method [1], with weak connection to the core equivariance theory. This paper has restated the highly related work done by Burgert et al. [2], however, there is no comparison with it.

[1] One-step Diffusion with Distribution Matching Distillation: https://arxiv.org/abs/2311.18828

[2] Go-with-the-Flow: Motion-Controllable Video Diffusion Models Using Real-Time Warped Noise : https://arxiv.org/pdf/2501.08331

**Strengths:**

1. Theoretical Clarity. Theorem 4.1 seems the most solid part of this paper. It clearly proves that under the use of warped noise and the standard denoising loss, the optimal solution naturally possesses equivariance. But there still exists some puzzles, in Eq.4, the authors utilize $\sum_k$, so why $\mathbf{V_t}$ is not written as $V^{(k)} + n^{(0)}$ ?

2. The visualization results seem good.

**Weaknesses:**

1. Lack of novelty. The core component, warped noise, is entirely derived from the work from others. This paper merely applies this existing method to the traininng of video diffusion models and proves a related theoretical property. This is a direct application, not a fundamental methodological innovation.

2. Missing comparison with related work. The paper repeatedly mentions the work of Burget et al. (Go-with-the-Flow), which also fine-tunes models using warped noise to improve motion control. This is a highly related and directly competitive work. However, in all experiments, this no comparison with this work. This severely weakens the paper's persuasiveness.

3. Vague method description, lack of reproducibility. This paper provides no pseudo-code, and I can not understand how the authors extracct the optical-flow, and how to use this to construct the wraped noise, and most critically, where is the definition of warping transfromation $\mathcal{T}_k$, how to calculate it? These implementation details are omitted, making the method nearly impossible to reproduce.

4. Is the paper's contribution proposing a new training framework, or merely proving a theoretical property? The content suggests the latter. Hope the authors can clarity the contribution of this paper.

**Questions:**

See the weakness.

---

> ### Author Response · Authors · 2025-11-21
> **Response to Reviewer T9uo**
>
> We thank the reviewer for recognizing the theoretical clarity of our work and the strength of our visualization results. We appreciate the opportunity to clarify our contributions regarding novelty, implementation and the theoretical notation.
>
> **1.Novelty, theoretical contribution and connection to distillation**
>
> We respectfully clarify that our contribution is not the invention of warped noise itself, but the discovery of its theoretical implications for Video Diffusion Models (VDMs). Our primary contribution is Theorem 4.1, which proves that training with warped noise using the standard denoising objective implicitly forces the denoiser to become equivariant to the spatial warping. This is a fundamental property that explains why warped noise works for temporal consistency, moving beyond heuristic application.
>
> The connection between our distillation method and the core theory is strong. As shown in Figure 4 and Section 5.3, the induced equivariance results in significantly straighter generation trajectories compared to independent noise models. This straightness is the critical factor that makes one-step distillation (DMD) highly effective for our model (FID 25.94) compared to baselines (FID 29.42). We explicitly draw this connection in Section 4.3
>
> In summary, we characterize the emergent properties (equivariance and trajectory smoothness) of warped noise training and leverage them for superior sampling efficiency.
>
> **2.Comparison with related work (Burgert et al.)**
>
> We cite the work from Burgert et al (“Go-with-the-Flow”) as highly relevant work. However, our contributions are distinct and complementary: Burgert et al. focus on "Motion Controllability" and user-directed outcomes. Our work focuses on Equivariance Theory and Sampling Efficiency; In addition, we go beyond standard generation to demonstrate Few-Step Generation and One-Step Distillation. While Burgert et al. show that finetuning helps motion control, they do not analyze the equivariance, the resulting trajectory curvature, or leverage the method for distillation.
> In a sense, our experimental results on the "base" EquiVDM validate the same underlying technique as Burgert et al. However, our Table 3 provides unique insights into sampling acceleration that are absent in their work. We will add a more explicit discussion in the related work section, clarifying these distinctions
>
> **3.Implementation details and reproducibility**
>
> We extract optical flow using RAFT. For the warping transformation $\mathcal{T}$, we adopt the Integral Noise formulation from Chang et al. (2024) , as stated in Section 3. This method aggregates deformed pixels from an upsampled noise image to preserve Gaussianity.
>
> Pseudo-code: We will add pseudo-code in the final revision. The process is: (1) Extract flow from video $V$; (2) Generate noise $n^{(0)}$; (3) Warp $n^{(0)}$ via RAFT flow using Integral Warping to get $n^{(k)}$; (4) Train VDM with standard loss on $V + n$.
>
> **4.Mathematical notation in Equation 4**
>
> We use $\mathbf{V}_t$ to denote a video (i.e., a sequence of frames) for concise notation. As stated in L136 and L184,
> $\mathbf{V}_t = \big(V^{(0)} + n^{(0)}, \ldots, V^{(k)} + n^{(k)}\big)$,
> where $V^{(k)}$ is the clean frame at time $k$ and $n^{(k)}$ is the corresponding noise for that frame. Both $V^{(k)}$ and $n^{(k)}$ represent single frames rather than full videos. Importantly, at frame $k$ we add the frame-specific noise $n^{(k)}$, not the noise $n^{(0)}$ from frame 0. The sequence $(n^{(0)}, \ldots, n^{(k)})$ is obtained by warping a base noise field over time to ensure temporal consistency across frames.
> In Eq. 4, the distribution of interest and the input to the denoiser is the entire diffused video $\mathbf{V}_t$, rather than an individual frame from that video.

---

> > ### Comment · Reviewer_T9uo · 2025-11-26
> > **Response to the authors**
> >
> > Thank you for your response. I have read your rebuttal carefully. However, my concerns regarding the **comparison with related work** and the **limited methodological novelty** remain largely unresolved.
> >
> > **1. Regarding the missing comparison with Burgert et al. ("Go-with-the-Flow"):**
> > I find the authors' justification for excluding a quantitative comparison with Burgert et al. unconvincing. You state that while the work is "highly relevant," the contributions are "distinct" because they focus on motion control while you focus on theory and distillation.
> >
> > However, since your method leverages the exact same core component (warped noise) as Burgert et al. to influence the generation process, **it is methodologically unreasonable to exclude it from your experimental baselines.** Even if the *downstream goals* differ (control vs. distillation), the underlying generative performance, temporal consistency, and FVD scores should be compared.
> > *   If your method is essentially a theoretical formalization of the technique used by Burgert et al., you must demonstrate empirically whether your theoretical constraints yield better, worse, or identical performance to their heuristic application.
> > *   Simply claiming a "different focus" does not exempt a paper from benchmarking against the most similar existing architecture. Without this comparison, it is impossible to verify if your specific implementation of warped noise offers any practical advantage over the existing state-of-the-art.
> >
> > **2. Regarding Novelty and Contribution:**
> > Your response confirms my initial concern: the core mechanism is not a new invention but an analysis of an existing one. You explicitly state, *"our contribution is not the invention of warped noise itself, but the discovery of its theoretical implications."*
> >
> > While I appreciate the derivation of Theorem 4.1, the paper is presented as proposing a new framework ("EquiVDM"). If the practical algorithm is merely a direct application of an existing method (warped noise) to standard training, the contribution is reduced to a post-hoc theoretical explanation of why previous methods work.
> >
> > A theoretical property alone, without a fundamental change in methodology or a demonstration of superior performance against the method it analyzes (Burgert et al.), represents a limited contribution. The argument that this theory facilitates distillation is valid, but since the core training paradigm remains derivative, the overall novelty is insufficient for a top-tier conference.

---

> > > ### Author Response · Authors · 2025-12-01
> > > **Response to Reviewer  T9uo**
> > >
> > > We thank the reviewer for their continued engagement and for clearly articulating their remaining concerns. We value the opportunity to provide a clarification regarding the comparison with related work and the nature of our methodological contribution.
> > >
> > > **Regarding Novelty and "Post-Hoc" Theory**
> > >
> > > We respectfully disagree with the characterization of our work as merely a "post-hoc theoretical explanation." A post-hoc analysis describes why a method works; our theoretical framework was theoretical and predictive, enabling new capabilities that were previously unseen.
> > > First, we theoretically derived that minimizing the denoising loss with warped noise implies equivariance. Crucially, we further demonstrated that this induced equivariance leads to smoother, straighter sampling trajectories (verified in Sec 5.3).
> > > Second, guided by the insight from the theoretical results, we introduced an efficient distillation method based on DMD and warped noise(Section 4.3): warped noise simplifies the ODE trajectory, making one-step distillation highly effective.
> > >
> > > In summary, the conclusion that warped noise induces equivariance and straightens trajectories for acceleration expands our understanding of video diffusion models and has not been presented in prior works. These are not "post-hoc explanations" of existing heuristics, but predictive theoretical insights that unlocked previously unrealized capabilities.
> > >
> > >
> > > **Regarding the Comparison with Burgert et al.**
> > >
> > > As noted in our previous response, our "base" EquiVDM effectively implements the same warped noise training paradigm utilized by Burgert et al. While the base training paradigm is similar, our work provides two distinct and fundamental contributions that extend beyond the scope of Burgert et al.
> > >
> > > (1) We proved in Theorem 4.1, that minimizing the standard denoising loss with warped noise implicitly forces the denoiser to become equivariant. This establishes a rigorous theoretical grounding for noise warping equivariance that is model-agnostic, irrespective of the specific VDM architecture or hyperparameter configurations
> > >
> > > (2) We leverage this theoretical insight to demonstrate that equivariant training straightens the generation trajectory, as shown in Figure 4. This discovery allows us to achieve high-fidelity results with significantly fewer sampling steps and enables effective one-step distillation (Table 3), a capability and advantage that is neither explored nor demonstrated in Burgert et al.
> > >
> > > We hope this clarifies that our contribution is the theoretical grounding that unlocks equivariance and sampling efficiency, rather than just the re-application of a specific noise warping operation or post-hoc explanation.

---

### Official Review · Reviewer_7Sgy · 2025-11-01

**Soundness:** 3
**Presentation:** 3
**Contribution:** 3
**Rating:** 6
**Confidence:** 5

**Summary:**

This paper proposes training video diffusion models with temporally warped noise to enforce inherent motion consistency and faster sampling. By replacing independent Gaussian noise with flow-warped noise during training, the model (termed EquiVDM) implicitly learns equivariance to spatial transformations of the noise, so that motion in the input noise translates into identical motion in the generated video without extra modules. The authors provide a theoretical proof (Theorem 4.1) that the standard denoising objective on warped noise yields an equivariant denoiser, aligning output frames with input motions. Empirically, EquiVDM achieves comparable or superior video quality with far fewer sampling steps, and a one-step distilled EquiVDM preserves this equivariance while improving motion fidelity over non-equivariant baselines. Comprehensive experiments across benchmarks demonstrate improved motion alignment, temporal coherence, and perceptual quality versus prior methods, supporting the paper’s claims

**Strengths:**

Theoretical insight: It provides a sound theoretical result – Theorem 4.1 – showing that using warped noise with the standard objective provably induces equivariance in the denoiser. This analysis is original and gives a clear explanation for why motion in input noise yields corresponding motion in outputs, solidifying the approach’s foundation.

Sampling efficiency: The study demonstrates faster sampling without quality loss. EquiVDM’s generation trajectory is much straighter (lower curvature) than the baseline’s, enabling high-fidelity videos in significantly fewer diffusion steps.

Comprehensive evaluation: The experiments are thorough and insightful. The authors evaluate both text-to-video and video-to-video generation scenarios with rigorous metrics , and include ablations (e.g. varying β in Eq. 5) to justify design choices. These ablations confirm that adding a small independent-noise component (β≈0.9) balances quality and consistency, lending credence to the approach’s robustness.

**Weaknesses:**

Dependence on optical flow: The approach relies on accurate motion vectors/optical flow from a driving video to warp noise, which limits applicability when such motion cues are unavailable (e.g. pure text-to-video generation). A suggested improvement is to incorporate a learned or inferred motion prior (e.g. a flow predictor from text prompts) so that EquiVDM can extend to scenarios without ground-truth flow.

Scope of equivariance theory: The theoretical equivariance guarantee assumes every frame is a perfect spatial warp of the first frame, an ideal scenario that may not hold for complex motions (e.g. new occlusions or content changes). It would strengthen the work to discuss how equivariance might be approximated when real motion deviates from pure warping, or to introduce training constraints that handle partial equivariance in more general settings.

Background consistency trade-off: In some evaluations, EquiVDM shows a slightly lower background consistency than the best baseline (e.g. minor drops in BgC metric). This suggests a trade-off between enforcing equivariance and preserving background fidelity. Please analyze these cases specifically and provide a targeted analysis in the rebuttal

**Questions:**

The evaluation relies on CLIP scores for both video-to-video and text-to-video tasks. However, CLIP was primarily standardized for image–text alignment and, even when averaged over frames, does not directly assess temporal coherence or frame-level consistency. To more faithfully measure textual adherence in videos, please complement CLIP with a dedicated video–text alignment metric (e.g., UMTScore [1]).  This addition would provide a stronger, temporally aware assessment beyond image-centric CLIP scores.

[1] Liu Y, Li L, Ren S, et al. Fetv: A benchmark for fine-grained evaluation of open-domain text-to-video generation[J]. Advances in Neural Information Processing Systems, 2023, 36: 62352-62387.

---

> ### Author Response · Authors · 2025-11-21
> **Response to Reviewer 7Sgy**
>
> We appreciate the reviewer’s recognition of our theoretical contributions (Theorem 4.1), the sampling efficiency of EquiVDM, and the comprehensive nature of our evaluation. We value the insightful feedback regarding the scope of our theory and metrics. We address your concerns below.
>
> **1. Dependence on optical flow**
>
> We fully agree with this assessment. As mentioned in the abstract, introduction, and conclusion sections, we explicitly positioned this work within the scope of video-to-video generation where driving videos are available. We agree that extending EquiVDM to text-to-video generation by using a learned flow predictor (e.g., generating optical flow from text prompts) is a promising direction. This would allow the model to leverage the benefits of equivariant warped noise without requiring a ground-truth driving video. We will expand the discussion in the revised manuscript to detail how such a module could be integrated.
>
> To further test the dependency on the accuracy of the optical flow, we conduct an experiment where the estimated optical flow vectors are deteriorated by random noise flow vectors. We sample the noise flow vectors from a Gaussian distribution in the same dimension as the optical flow. Then we run sample videos using the noisy optical flow. The evaluation results are shown below:
>
> | σ  | FID↓ | FVD↓ | CLIP↑ | cf-PSNR↑ |
> |----|------|------|-------|----------|
> | 2  | 23.95| 2812 | 0.7547| 26.73    |
> | 5  | 24.79| 2883 | 0.7529| 27.04    |
> | 10 | 25.15| 2955 | 0.7520| 26.28    |
> Table. 1 Performance of EquiVDM vs. the amount of optical flow noise added before noise warping.
>
> As shown, our method’s performance degrades gracefully even for large amounts of optical flow noise (e.g. sigma=10).
>
> **2. Scope of equivariance theory (Occlusions and non-overlapping regions)**
>
> This is an insightful observation. We will refine the scope of the theorem to account for these real-world deviations without invalidating the core result.
> For theoretical refinement: To formally address this, we can restrict the transformation $\mathcal{T}$ to the visible spatial overlap between frame $0$ and frame $K$. Under this restriction, the conclusions of Theorem 4.1 hold strictly for the overlapping regions where the mapping is bijective and the denoising is equivariant. We do not make theoretical claims for non-overlapping (occluded) regions, where in practice the model “hallucinates” new pixels.
> Regarding practical Handling: In our implementation, we address the deviation from pure warping by detecting disoccluded or new regions during the noise generation process and filling these areas with independent Gaussian noise. This ensures that the model relies on equivariance where valid (the overlap) and falls back to standard generation where the warping assumption breaks. We will clarify this distinction in the revised theory section.
>
> **3. Background consistency score drop**
>
> Background Consistency (BgC), defined in the VBench benchmark, calculates the cosine similarity of CLIP features between consecutive frames and the first frame. As a result, it inherently favors static backgrounds. A model that ignores motion cues and generates a "frozen" video will achieve a higher BgC score, even if that contradicts the driving video's motion.
>
> On the other hand, EquiVDM is trained to be equivariant to the input optical flow. When the driving video contains camera motion (e.g., pans, zooms), EquiVDM faithfully renders these perspective shifts in the generated background. While this is the wanted behavior for video-to-video translation, it naturally results in lower CLIP similarity scores compared to baselines that fail to capture the motion and default to static backgrounds. The minor drop in BgC is not a quality degradation but a reflection of motion fidelity. This interpretation is supported by our significantly higher Motion Alignment (cf-PSNR) and Subject Consistency (SubC) scores, which confirm that EquiVDM is superior in preserving the actual dynamics of the scene.
>
> **4.UMT-score**
>
> We thank the reviewer for this valuable suggestion. We have added UMTScore to our evaluation framework in the revised PDF. The results confirm that EquiVDM maintains strong semantic alignment even when evaluated with a video-native metric like UMT, which is better suited to capturing temporal dynamics than frame-averaged CLIP. This further substantiates that our method achieves motion consistency without sacrificing semantic textual adherence.

---

> > ### Comment · Reviewer_7Sgy · 2025-11-26
> >
> > Thank you for the detailed  rebuttal. It has addressed my main concerns and clarified the issues I raised. I am satisfied with the authors’ responses and will keep my positive rating for this paper.

---

### Official Review · Reviewer_yJdz · 2025-11-05

**Soundness:** 2
**Presentation:** 3
**Contribution:** 2
**Rating:** 2
**Confidence:** 4

**Summary:**

This paper fine-tunes a pretrained video diffusion model using *warped (flow-correlated) noise* instead of i.i.d. Gaussian noise, claiming this induces equivariance and enables faster sampling. Experiments show improved motion alignment and fewer sampling steps.

**Strengths:**

1. The paper demonstrates that simple warping to the noise distribution yields measurable quality enhancement and speed-up.

2. The experiment results demonstrate improved temporal consistency when optical flow is available.

3. The implementation is compatible with existing video diffusion architectures.

**Weaknesses:**

1. **The paper overlooks the gap between the theoretical result and the behavior of real diffusion models.**

    Theorem 4.1 characterizes the Bayesian optimal denoiser under highly idealized assumptions (perfect warping, linear transform, noise consistency, no occlusion, etc.).
    In practice, current video diffusion models do not contain any mechanism that enforces or even approximates $D_\\theta^{(k)}(V_t) = T_k \\circ D_\\theta^{(0)}(V_t)$, either in pixel space or latent space.
    As a result, the model can trivially minimize the loss by memorizing and locally denoising the warped noise, instead of learning true equivariant behavior.
    To substantiate the claim of “emergent equivariance,” I suggest that the authors directly test equivariance on synthetic warp operators (i.e., warp the noise, feed it to the model, and check whether the output matches the warped baseline).
    From my view, evaluating only motion alignment in generated frames may not validate the equivariance of the denoiser itself.

2. **The practical constraints of the method outweigh its claimed benefits.**

    In my opinion, the model is likely to merely restrict the source distribution by injecting external motion information into the noise, rather than improve the denoiser’s intrinsic motion modeling ability.
    In fact, a strong Gaussian denoiser is fine-tuned into a weaker, conditional denoiser that only works on a specially correlated noise distribution.
    This intuitively weakens the model's generation capability of diverse/unseen motion.
    To verify this, the authors should report performance under *OOD flow fields* or randomized flow noise.

3. **The method doesn't provide significant advantages over existing optical-flow-based guidance.**

   Unlike inference-time conditioning or plug-and-play guidance, the proposed modification permanently alters the model: It can no longer sample from standard i.i.d. Gaussian noise, nor function without optical flow.
    The paper acknowledges this only briefly in the limitations section, but does not quantify the severity (e.g., failure modes when flow is unavailable, inaccurate, or out-of-domain).
    No evaluation is provided on pure Gaussian noise, erroneous flow, or non-warpable videos, which undermines the generality of the proposed method.

**Questions:**

Please refer to the weaknesses.

---

> ### Author Response · Authors · 2025-11-21
> **Response to Reviewer yJdz**
>
> We thank the reviewer for acknowledging that our method yields measurable quality enhancements, improves temporal consistency, and is compatible with existing architectures. We respectfully disagree with the assessment that the model fails to learn equivariance and merely "memorizes" noise. We address these concerns below by clarifying our evaluation metrics and the scope of our contribution.
>
> **1.Current video diffusion models do not contain any mechanism that enforces or even approximates equivariance, either in pixel space or latent space. The model trivially minimizes loss by memorizing and "locally denoising" warped noise rather than learning true equivariance.**
>
> We fully agree with this observation that the current models do not contain mechanisms to enforce equivariance. The lack of intrinsic equivariance in standard models is precisely the problem our work addresses, both theoretically and empirically.  We emphasize that our approach is not an assumption that models are equivariant, but a method to induce equivariance. Theorem 4.1 proves that minimizing the standard denoising loss with warped noise optimizes the denoiser towards equivariance. Our experiments confirm this optimization succeeds in practice.
>
> As for the suggested validation experiment to address the concern around memorization and local denoising, we respectfully point out that we have already performed the quantitative test similar to what the reviewer suggests, which we report as cf-PSNR. As described in Section 5 and Appendix A, cf-PSNR measures precisely whether the generated video follows the transformation $\mathcal{T}$ applied to the input noise. Specifically, we generate a frame from warped noise ($D(\mathcal{T}(n))$) and compare it to the previous generated frame warped by the same flow ($\mathcal{T}(D(n))$). A high cf-PSNR indicates that $D(\mathcal{T}(n)) \approx \mathcal{T}(D(n))$, satisfying the definition of equivariance.
>
> On the other hand, if the model were merely performing local denoising (treating warped noise as independent noise patterns without global awareness), the generated frames would lack semantic and structural temporal coherence. However, our results in Tab.2 show that EquiVDM achieves superior semantic consistency and frame quality. This proves the model is not simply memorizing noise patterns but preserving semantic integrity across the frames.
>
> **2.The method weakens the model's generation capability of diverse/unseen motion.**
>
> In V2V applications (e.g., style transfer, simulation), the goal is not to generate random diverse motion, but to strictly adhere to the motion of the driving video. By constraining the noise distribution to follow the input flow, we provide a strong prior that ensures high-fidelity motion alignment.
>
> In addition, regarding the concern that the model relies on memorized patterns or is restricted to specific training motions, we emphasize that our evaluation is performed on datasets entirely distinct from our training data. We trained EquiVDM on RealEstate-10k, OpenVideo-1M, and VidGen-1M, but evaluated on the separate Youtube-VIS 2021 and MSRVTT benchmarks. The model’s strong performance on these test sets—which contain diverse, in-the-wild motion dynamics not seen during training—confirms that EquiVDM generalizes effectively to unseen motion fields rather than merely overfitting to a restricted source distribution.
>
> As suggested, we validated the robustness of the model against random flow noise - please refer to Tab.1 and its discussion in the response to Reviewer 7Sgy.
>
> **3.The method offers no significant advantage over inference-time conditioning and optical-flow-based guidance methods**
>
> We believe EquiVDM offers three distinct, critical advantages over inference-time guidance (e.g., Warped Diffusion)
> Efficiency (Few-Step Sampling):
>
> 1) Inference-time guidance requires calculating gradients through the network at every sampling step, which is computationally expensive. In contrast, EquiVDM learns the equivariance into the model . This allows for extremely fast sampling. As shown in Figure 8 and Table 3, EquiVDM maintains high quality even at 5 steps, whereas standard models degrade significantly.
>
> 2) Better few-step distillation: Unlike inference-time guidance, which guides the ODE trajectory with external gradient terms, our method naturally aligns the input noise motion with the output video motion. As visualized in Figure 4 and discussed in Section 5.3, this alignment significantly lowers the curvature of the generation trajectory compared to models trained with independent noise. This inherent "straightness" simplifies the distillation task. Consequently, we are able to distill EquiVDM into a high-fidelity one-step model as validated in Tab3.
>
> 3) Simplicity: Compared to the inference-time guidance methods, we achieve state-of-the-art temporal consistency without complex auxiliary loss functions, hyperparameter tuning during inference, or additional control modules.

---

> > ### Comment · Reviewer_yJdz · 2025-11-28
> >
> > I thank the authors for their detailed response.
> > I have carefully read the rebuttal from the authors and the comments from other reviewers.
> > While the authors address part of my concerns on the theoretical aspects, I incline to agree with Reviewer T9uo on the concerns related to the contributions. Thus, I would raise my score from 2 to 4 accordingly.

---

> > > ### Author Response · Authors · 2025-11-29
> > >
> > > We’re glad to hear that we addressed part of your concerns, and appreciate that you raised your score positively from 2 to 4 on Nov. 28th. Thank you for your feedback!

---

### Official Review · Reviewer_Rqsb · 2025-11-06

**Soundness:** 3
**Presentation:** 3
**Contribution:** 3
**Rating:** 6
**Confidence:** 5

**Summary:**

This paper addresses temporal consistency and sampling efficiency in video diffusion models, focusing on the effects of using warped noise during training. The authors theoretically prove (Theorem 4.1) that training with temporally warped noise induces equivariance to spatial transformations without the need for architectural modifications or auxiliary losses. They introduce EquiVDM, an equivariant video diffusion model, and propose a distribution-matching distillation method that trains fast, one-step student models while preserving motion controllability and fidelity. The approach is comprehensively evaluated and achieves compelling improvements in motion alignment, temporal coherence, and sample efficiency compared to baselines and prior state-of-the-art methods.

**Strengths:**

Theoretical Depth and Clarity: The paper delivers a clear theoretical foundation, most notably Theorem 4.1, which formally establishes that training diffusion models with warped noise enforces equivariance to warping transformations under the standard objective. The proof is sound and leverages concise, well-articulated mathematical reasoning (see Section 4.1).

Practical Applicability: Beyond theory, the paper details a straightforward recipe for integrating warped noise training into off-the-shelf state-of-the-art video diffusion architectures, including both UNet and transformer-based backbones. It is notable that no new architectural components or loss functions are introduced—a rarity for claims of improved temporal coherence.

Significant Empirical Results: Across several large-scale datasets (OpenVideo-1M, VidGen-1M, Youtube-VIS, MSRVTT), the empirical evaluation is thorough and multidimensional, covering FID, FVD, CLIP, cf-PSNR, ImQ, BgC, SubC, and others. In Table 1 and Table 2, EquiVDM consistently and substantially outperforms strong baselines such as VC2, Show-1, and state-of-the-art video control methods on both quality and temporal consistency metrics.

Fast Sampling/Distillation: The proposed distribution-matching distillation (DMD) enables one-step inference while retaining competitive performance (see Table 3 and Figure 8), with substantial quality preservation compared to traditional multi-step sampling. This is a substantial practical advantage.

**Weaknesses:**

The core approach requires high-quality motion vectors from optical flow estimation for every video, as outlined in Section 4.1 and acknowledged in the conclusion. This dependence on a non-trivial, error-prone subroutine introduces fragility: as observed in the paper (Section 4.2, Figure 2), errors or occlusions in optical flow estimation can break the required equivariance, especially in complex scenes with large motion or occlusions.

The authors themselves note—and Figure 2 demonstrates—that applying warped noise in latent space does not always yield the theoretically desired temporal consistency, due to unpredictable mappings from pixel to latent space and high-frequency encoder artifacts (Section 4.2). Although the proposed solution of injecting a small amount of independent noise (Equation 5) is empirically validated (Table 4), there is no theory provided to justify the optimality or generality of this heuristic. The tradeoff between motion consistency and noise manifold coverage remains a “hack.”

Theorem 4.1, while sound, makes idealized assumptions (all frames perfectly correspond under warping transformations and noise is perfectly warped). The paper acknowledges practical violations due to encoder drift (Section 4.2), occlusions, and imperfect flow, which dilutes the theorem’s generality. There is little quantitative measurement of how far real-world data/processes deviate from the theorem’s assumptions, and the model’s equivariance is not characterized beyond qualitative results.

Potential Drifting in Long Videos: As noted in the conclusion, introducing only warped noise is not sufficient to fully eliminate temporal drift over long sequences. No experiments are reported for video lengths well beyond those seen in training, and the issue of maintaining long-range consistency is largely unexplored. This is a significant omission, given the practical importance of long video synthesis.

Abrupt Handling of Non-Available Motion Priors: The method depends on motion priors extracted from input (“driving”) videos, which are unavailable in text-to-video or generative settings without external cues. The discussion around synthesizing plausible flow from text prompts is surface-level (Section 6), and the approach is currently limited to video-to-video scenarios.

**Questions:**

See weaknesses.

---

> ### Author Response · Authors · 2025-11-21
> **Response to Reviewer Rqsb**
>
> We thank the reviewer for their positive assessment and for highlighting the strengths of our work, particularly the theoretical depth of Theorem 4.1, the practical applicability of the method, and the strong empirical results in sampling efficiency. We value constructive feedback regarding the assumptions of our theorem and the method's scope. We address these points below.
>
> **The "Heuristic" nature of latent space noise**
>
> While Equation 5 is empirical, it is grounded in the observation that latent encoders are not equivariant (Section 4.2 and Figure 2). The independent noise component expands the noise manifold, allowing the model to cover the high-frequency variances in the latent space that the warped noise (which lies on a lower-dimensional manifold) cannot capture. We view this not as a "hack," but as a necessary adaptation to apply theoretical equivariance (which holds in signal space) to practical Latent Diffusion Models (LDMs). The empirical success in Table 4 validates this design choice
>
> **There is little quantitative measurement of how far real-world data/processes deviate from the theorem’s assumptions**
>
> We acknowledge that Theorem 4.1 assumes an ideal scenario. However, our method and experimental setup effectively test how well the model performs when these assumptions are relaxed (i.e., when using imperfect, real-world optical flow).
>
> Method-wise: As discussed in Section 4.2 and illustrated in Figure 2, we explicitly address the deviation between pixel-space warping and latent-space dynamics. We introduce the mixed noise formulation $n = \beta n_{warp} + \sqrt{1-\beta^2} n_{ind}$ (Equation 5) specifically to handle the high-frequency fluctuations and flow inaccuracies that the reviewer mentions. The quantitative measurements in ablation (Table 4) show that pure warped noise ($\beta=1.0$) does indeed degrade, but our proposed mixture ($\beta=0.9$) achieves optimal performance (FID 25.12), significantly outperforming the baseline.
>
> Experiment-wise: Furthermore, we emphasize that all our results utilize off-the-shelf optical flow estimators (RAFT), which inherently contain errors and occlusions. The fact that EquiVDM consistently outperforms baselines in Table 2 demonstrates that the model is robust to these standard deviations from the "idealized" theorem assumptions
>
> To further demonstrate this, we conduct an experiment where the estimated optical flow vectors are deteriorated by random noise flow vectors. Please refer to Tab.1 in the response to Reviewer 7Sgy.
>
> **The model’s equivariance is not characterized beyond qualitative results**
>
> We respectfully clarify that we do provide a direct quantitative measure of equivariance via the cf-PSNR (Cross-Frame PSNR) metric. As detailed in Section 5.1 and Appendix A (Figure 7), cf-PSNR measures the alignment between the motion in the generated video and the motion vectors of the input driving video. Theoretically, if a model is perfectly equivariant, the generated frames should follow the exact same warping transformation $\mathcal{T}$ applied to the input noise. Therefore, a higher cf-PSNR score directly indicates a higher degree of learned equivariance.
>
> As shown in Table 2, EquiVDM achieves significantly higher cf-PSNR scores than the baselines, quantitatively proving that our model has learned to be more equivariant to the input flow than the baselines.
>
> **Abrupt handling of non-available motion priors**
>
> We wish to clarify the scope of our contribution. As stated in the Abstract and Introduction, EquiVDM is explicitly designed for video-to-video generation tasks (e.g., style transfer, upsampling, simulation). We do not claim it is a T2V generation model. In addition, the limitation regarding T2V is acknowledged in Section 6, but we believe the substantial improvements in the V2V domain stand as a significant contribution on their own.
>
> **Potential drifting in long videos**
>
> We agree that autoregressive drift is a challenge for diffusion-based video generation. We explicitly identified this as a limitation in Section 6. Our primary contribution addresses motion fidelity and sampling efficiency within standard generation windows. We have pointed to future directions, such as integrating EquiVDM with autoregressive approaches like Diffusion Forcing or Self Forcing

---

### Author Response · Authors · 2025-11-21
**Message to all reviewers and ACs**

This paper is an attempt to understand what would happen if a video diffusion model is trained with warped noise. We theoretically and empirically show that under certain mild assumptions, video diffusion models with warped noise will be trained to be equivariant. We also empirically show that these models have smoother sampling trajectories and can be accelerated easily by reducing the number of sampling steps or distillation. These novel conclusions expand our understanding of video models, are of significant interest to the community, and have never been presented in any prior works.

We are frankly disheartened that the reviewers do not value the conclusions and criticize the paper for minor issues. Below, we address all the concerns raised by the reviewers. We hope that the reviewers will consider raising their scores.

---

### Meta-Review · Area_Chair_RXtf · 2026-01-06

**Summary:**

The paper is about video generation with the warped noise, i.e. warping the noise of diffusion model according to optical flow of the video. The paper provide some theoretical insights that such a model is leaning towards equivariant solutions. The authors also demonstrate that the model with warped noise is better at few steps inference.

The reviewer recommendations was divided 6,6,4,4. With main non-addressed concerns being: limited scope of applications, reliance on optical flow routine and idealized theoretical assumptions.

Area chair believe these concerns especially limited scope of application to be major and recommends rejection of the paper. Mainly the model can be applied only to a subset of V2V tasks, where optical flow is the same between source and target videos (depth-2-video, edge-2-video or style transfer not affecting the video structure), v2v task where structure modification is needed (for example changing motion of one of the objects, changing camera or style transfer with large structural changes natural video 2 cartoon) is not tested and does not align with the theoretical justifications.

**Reviewer Concerns:**

*Rqsb* mainly has two concerns related to inaccuracies and limitations of optical flow and over idealized assumption of theorem 4.1. Area chair believes limitations of optical flow concern was not addressed, optical flow warping reduce applicability of the method for long video and videos with scene changes even in V2V settings. Area chair however find idealized assumption to be needed to provide theoretical insights, and being reasonably addressed during the discussion.


*yJdz* has 3 concerns:
1) Idealized assumption for theorem 4.1, area chair believe this concern to be reasonably well addressed.

2) Constraints of the method outweighs the benefit, area chair agree with this, since the method heavily limit the applicability to only specific V2V tasks.

3) No advantage over optical flow guidance, area chair believe this concern was addressed during the dissussion.

*7Sgy* has 3 concerns but authors responses was satisfactory for him.

*T9uo* has 3 concerns:

1) Lack of novelty, unclear scope of contribution. Based on the discussion reviewer was not convinced by author response.

2) No comparison to the baseline. Authors justification was that baseline utilize the same core mechanism as their Go-with-the-Flow, and their method provide more insights into this approach. So this point is related to 1) Lack of novelty, of which *T9uo* was not convinced.

3) Unclear Method description. Area chair believe this point to be reasonably well adressed.

**Reviewer Scores:**

Rqsb - stay the same

yJdz - increase to 4

7Sgy - stay the same

T9uo - stay the same

---

### Decision · Program_Chairs · 2026-01-26

Reject